

# Spatiotemporal variability of flash floods and their human impacts in the Czech Republic during the 2001–2023 period

Rudolf Brázdil[1,2], Dominika Faturová[1], Monika Šulc Michalková[1], Jan Řehoř[1,2], Martin Caletka[3], Pavel Zahradníček[2,4]

[1]Institute of Geography, Masaryk University, Brno, 611 37, Czech Republic
[2]Global Change Research Institute of the Czech Academy of Sciences, Brno, 603 00, Czech Republic
[3]T. G. Masaryk Water Research Institute, Brno, 612 00, Czech Republic
[4]Czech Hydrometeorological Institute, Brno, 616 67, Czech Republic

*Correspondence to*: Rudolf Brázdil (brazdil@sci.muni.cz)

**Abstract.** Flash floods, characterized by their sudden onset, extreme discharges, short duration, material damage, and human loss, represent a significant natural hazard. Not well covered by standard hydrological observations, flash floods data can

primarily be derived from various types of documentary evidence. This evidence served as the main data source for creating a flash flood database for the Czech Republic from 2001 to 2023. This database enabled detailed analysis of different aspects of flash floods. The annual series of the numbers of flash flood events, flash flood days, and affected municipalities showed significant inter-annual variability but no linear trends. The triggering rainfalls that generate flash floods were analyzed with respect to 1–3-hourly and daily precipitation totals and circulation types from the objective classification. While flash floods

can occur anywhere, they were more frequently recorded at the foots of mountain slopes, often coinciding with "critical points" where built-up areas meet concentrated surface runoff pathways. The division of material damage caused by flash floods into eight categories indicated that the highest proportions of damage were to streets and communications, as well as to houses, their cellars, and basements. There were also several recorded fatalities. The understanding of flash floods in the Czech Republic aligns generally well with studies of flash floods in other European regions.

**1 Introduction**

A flash flood is generally a situation in which water in a watercourse suddenly rises and overflows its banks following a relatively short period of torrential rain, often causing significant material damage and human loss (Archer and Fowler, 2016). Flash floods can also be described as strong flows that occur shortly after rainfall (Gruntfest and Huber, 1991) or as one of the major natural hazards in small streams (Bačová Mitková et al., 2018). Kaiser et al. (2021) differentiated flash floods and pluvial

floods, both triggered by heavy precipitation but exhibiting different behaviors. Flash floods are initiated by short-duration, high-intensity rainfall, leading to a rapid water torrent, while pluvial flooding results from water flowing towards watercourses, often independent of watercourses (Kaiser et al., 2021). Research on flash floods heavily depends on historical event data (*e.g.*, Archer et al., 2016; Kaiser et al., 2020).



A broad range of flash flood analyses from various perspectives appears at the international level. Studies include

analyses of individual historical events (*e.g.*, Llasat et al., 2003; Gaume et al., 2004; Thorndycraft et al., 2006; Borga et al., 2007; Braud et al., 2010; Ruiz-Bellet et al., 2015; Papagiannaki et al., 2017; Diakakis et al., 2019; Pekárová et al., 2021), national-scale spatiotemporal variability (*e.g.*, Bhaskar et al., 2000 for Eastern Kentucky; Gourley et al., 2013 for the U.S.; Bryndal, 2015 and Bryndal et al., 2017 for Poland; Trobec, 2017 for Slovenia; Archer et al., 2019 for England; Kaiser et al., 2021 for Germany), regional studies (*e.g.*, Llasat et al., 2010; Petrucci et al., 2012; Amponsah et al., 2018), and even continental

analyses (*e.g.*, Gaume et al., 2009; Marchi et al., 2010 for Europe). The relationship between flash floods and (convective) precipitation was explored in regions like Catalonia, Spain (Llasat et al., 2014, 2016). Atmospheric conditions conducive to extreme rainfall and the genesis of flash floods in central western Europe were investigated by Meyer et al. (2022). Research has also addressed physical-geographic (basin) and other factors contributing to the onset of flash floods (*e.g.*, Grešková, 2005; Minea, 2013; Borga et al., 2014; Bryndal, 2014; Braud et al., 2016; Zeleňáková et al., 2016; Saharia et al., 2017; Costache and

Tien Bui, 2020). Post-event damage assessments have led to the development of different indices like the Flash Flood Severity Index (FFSI) by Schroeder et al. (2016) or the Flash Flood Potential Index (FFPI) and the Flash Flood Residential Hazard (FFRH) by Shehata and Mizunaga (2018). The topic of fatalities related to flash floods has also been a focus of study (*e.g.*, Sharif et al., 2012; Vinet et al., 2016, 2022; Terti et al., 2017; Ahmadalipour and Moradkhani, 2019; Diakakis et al., 2023). Despite this, flash flood research continues to face challenges related to data scarcity (Kaiser et al., 2020) and database

incompatibility at the international level.

Concerning the Czech Republic, the analysis of flash floods has particularly focused on individual events from the pre-instrumental period, based on documentary data (*e.g.*, Munzar, 2003; Elleder et al., 2014), as well as on events during the period covered by systematic meteorological and hydrological measurements (*e.g.*, Balatka and Sládek, 1980; Chamas and Kakos, 1988; Polišenský, 1990; Hančarová et al., 1999; Cyroň and Kotrnec, 2000; Soukalová, 2002; Šimandl et al., 2007;

Šálek and Kaplická, 2008; Lipina et al., 2016). Detailed descriptions of several flash floods that occurred in June and July 2009 were provided by Kubát (2009), and for August 2010 by Kubát (2010). Raška and Brázdil (2015) used documentary sources to reconstruct social responses to five disastrous flash floods between 1897 and 1927 in northern Bohemia. Štěpánková et al. (2017) focused on the assessment of flash flood hazards in urbanized areas. Halásová and Brázdil (2020) presented a comprehensive spatiotemporal study of flash floods in the eastern part of the Czech Republic (Moravia and Silesia) for the

19th–20th centuries. Spálovský et al. (2022) investigated the hydrodynamic characteristics of surface runoff generated by flash floods in geologically diverse areas of the Czech Republic. Sercl et al. (2023) developed an operational approach to determining the real-time risk of flash flood occurrence in the Czech Republic, utilizing the current potential of rainfall-runoff modeling.

The official definition of a flood in the Czech Republic as presented in the Water Law does not specifically mention

flash floods: "Flood means a temporary significant increase in the water level in watercourses or other surface waters, during which the water overflows the watercourse bed and can cause damage. A flood is also a condition when water can cause damage by temporarily not draining from a certain area naturally or when drainage is insufficient, or when an area is flooded

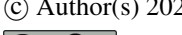



due to concentrated outflow of precipitation" (Act No. 254/2001/§64(1), Water Law). According to Daňhelka et al. (2015), a flash flood is characterized by a rapid rise in water level lasting from minutes to a few hours and a significant impact of the

dynamic force of turbulent flow as a factor in the occurrence of flood damage.

Similar to Kaiser et al. (2021), our study primarily considers flash floods directly connected to a watercourse, excluding cases associated with torrential rain causing surface runoff from fields, slopes, or streets in settlements. The aim of this study is to conduct a comprehensive analysis of flash floods in the Czech Republic during the 2001–2023 period, based on a newly created unique database of these events. Our research focuses on the spatiotemporal variability of flash floods, and

the meteorological, hydrological, and geographic factors contributing to their occurrence and progression, as well as their consequences – material damages and human losses. At the same time, the study addresses a gap in the broader international knowledge of Czech flash floods, which have not been included in related central European or pan-European studies (*e.g.*, Gaume et al., 2009; Marchi et al., 2010; Amponsach et al., 2018).

## 2 Data

### 2.1 Data on flash floods

Due to the typically small local and regional spatial extent of flash floods (hereafter referred to as FFs), they may not always be captured by systematic observations in the network of meteorological and hydrological stations of the Czech Hydrometeorological Institute (CHMI). The challenge of creating a comprehensive dataset of such events is underscored by the fact that, apart from the study by Halásová and Brázdil (2020) for Moravia and Silesia, which concludes in 1999, no recent

systematic database exists in the Czech Republic (CR). To gather data on FFs, various sources have been utilized, focusing on those reporting FFs from 2001 to the present, as described below.

### 2.1.1 Newspapers

Newspapers frequently provide detailed reports on hydrometeorological events, including FFs, describing the resulting material damage and loss of human lives. We systematically used the printed newspaper *Právo* and its online version

*Novinky.cz*, scanning them using specific keywords. For instance, *Právo* (1 June 2005, p. 7) detailed an FF on the Dubanka rivulet at Rozhovice (for locations of reported places in the CR see Fig. 1): "This [Dubanka rivulet] on Monday [30 May 2005] in the evening, during a severe storm, overflowed its banks within minutes, flooding nearly thirty houses and business premises. The rivulet rose by nearly two meters ... Water tore away a section of the road, inundated the lower part of the village with mud, but the situation was worse in houses inundated by water. 28 houses were flooded, with 12 of them also having their

living spaces damaged. Five families were forced to evacuate ... In [the village] Čepí, the damage was less severe than in neighboring Rozhovice, yet a well-maintained tennis court was buried under mud, water from six wells became unusable for a long time, and the scars of the rivulet's fury will mark gardens and fences for an extended period." Detailed reports from





*Právo* were supplemented by articles from other national newspapers (*e.g., Lidové noviny, Mladá Fronta Dnes*) and regionally focused publications (*e.g., Rovnost*).

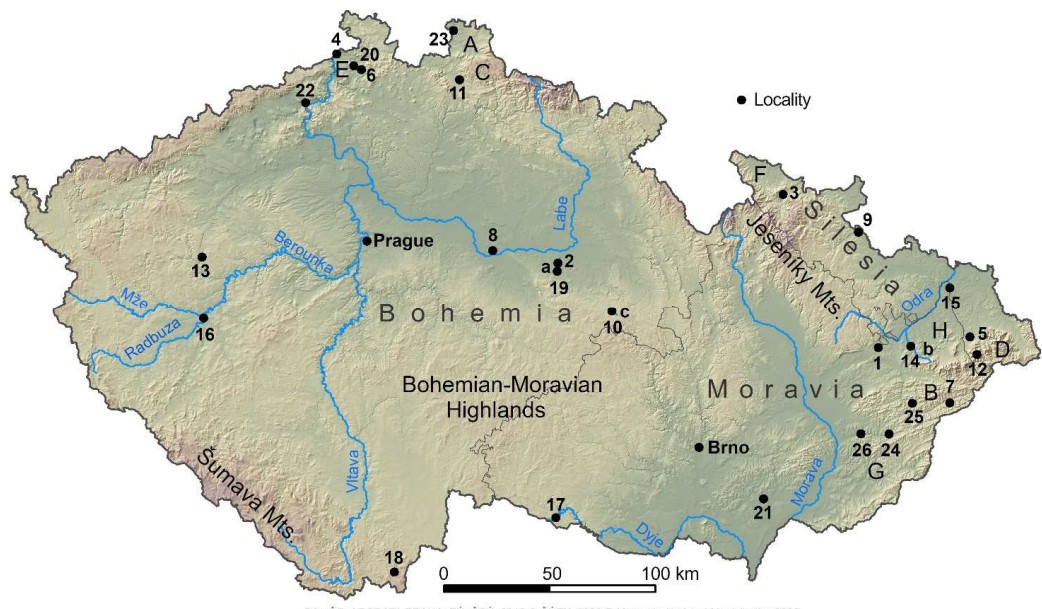


**Figure 1: Location of places, watercourses and geomorphological units in the Czech Republic mentioned in the text. Localities: 1 – Bělotín, 2 – Čepí, 3 – Česká Ves, 4 – Hřensko, 5 – Janovice, 6 – Janská, 7 – Karolinka, 8 – Konárovice, 9 – Krnov, 10 – Krouna (Rychnov), 11 – Liberec, 12 – Lysá hora, 13 – Mladotice, 14 – Nový Jičín, 15 – Ostrava, 16 – Plzeň, 17 – Podhradí nad Dyjí, 18 – Pohorská Ves, 19 – Rozhovice, 20 – Srbská Kamenice, 21 – Šardice, 22 – Ústí nad Labem, 23 – Višňová, 24 – Vizovice, 25 – Vsetín,**
**26 – Zlín. Watercourses: a – Dubanka, b – Jičínka, c – Rychnovský potok. Geomorphological units: A – Frýdlantská pahorkatina Hilly Land, B – Hostýnsko-Vsetínská hornatina Mts., C – Jizerské hory Mts., D – Moravian-Silesian Beskids, E – Labské pískovce Sandstones, F – Rychlebské hory Mts., G – Vizovická vrchovina Highlands, H – Západobeskydské podhůří Foothills.**

### 2.1.2 Internet sources

Searches using several FF-related keywords on the internet provided additional information not covered in newspapers. For
example, the website of Krouna municipality (Schmied, 2020) reported: "On Sunday, 14 June 2020, shortly after noon, the Rychnov area [a part of Krouna] was struck by a flash flood. In less than thirty minutes, large torrents of water flooded the Rychnovský potok Brook, which overflowed its banks and inflicted hundreds of thousands [of Czech crowns] in damage to settlement property and the possessions of Rychnov's citizens. The flood destroyed local roads, damaged small bridges, flooded cellars and wells, and ruined fences, gardens, and other property." This report was also supported by photos and videos.



### 2.1.3 CHMI reports

Several FFs were detailed in special reports from the CHMI, which typically provide extensive information on the related meteorological situations and precipitation totals, hydrological conditions, evaluation of weather forecasts, and other CHMI activities associated with the event. However, these reports usually focus less on human impacts. They also include instances where water levels in watercourses increased suddenly but did not overflow their banks; such cases were not included in our FF database.

### 2.1.4 Professional papers

Certain significant FFs or those of particular interest from meteorological or hydrological perspectives have been the subjects of specialized professional papers, sometimes building on the previously mentioned internal CHMI reports. Examples include studies of FFs on 15 July 2002 (Soukalová, 2002), 23 May 2005 (Šálek and Kaplická, 2008), 19 August 2007 (Šimandl et al., 2007), several events in June and July 2009 (Kubát, 2009), and in August 2010 (Kubát, 2010).

### 2.2 Other data sources

To analyze the climatology of rainfall triggering FFs in the CR from 2001 to 2023, precipitation measurements from CHMI stations were utilized. This was due to the limited availability of radar precipitation data for a portion of the analyzed period, although radar data would be more suitable for such analysis. For each FF event, corresponding 1-hour, 2-hour, 3-hour, and daily (24-hour) precipitation totals were collected, along with indications of thunderstorm occurrences at stations located in or near the affected areas. The station with the highest total was selected for further statistical analysis.

To characterize the synoptic conditions leading to the triggering rainfall totals for FF origin, an objective classification of circulation types was used. Following the methodology first introduced by Jenkinson and Collison (1977), this classification is based on the calculation of flow strength, flow direction, and vorticity from sea-level pressure in the ERA5 reanalysis (Hersbach et al., 2020) for the geographic center of the CR. Threshold values for these three parameters were used to distinguish three basic groups of circulation types, further divided according to eight basic airflow directions as follows: anticyclonic types – A, AN, ANE, AE, ASE, AS, ASW, AW, and ANW; cyclonic types – C, CN, CNE, CE, CSE, CS, CSW, CW, and CNW; and directional types – N, NE, E, SE, S, SW, W, and NW. Days that did not fit any of the preceding types were categorized as unclassified type U (for more details and sea-level pressure fields of individual types, see Řehoř et al., 2021a, 2021b).



## 3 Methods

### 3.1 Database of flash floods

Information on FFs in the CR from 2001 to 2023 extracted from the sources specified in Sect. 2.1 was compiled to create a corresponding FF database. For each FF event, the following data were collected:

(i) Year, month, day, part of the day, and exact time of the FF occurrence.

(ii) Affected municipality, part of the municipality, district.

(iii) Watercourse.

(iv) Triggering rainfall total.

(v) Maximum water level and maximum discharge rate.

(vi) Degree of flood activity.

(vii) Category of material damage: A – flooded cellar/basement, B – flooded house, C – flooded street/road, D – flooded garden, E – damaged road, F – other damage, G – landslide, H – non-specified damage.

(viii) Number of flood fatalities (direct or indirect).

(ix) Full original report.

(x) Source (including picture and video documentation).

Of course, not every detected FF provided complete information for all the points (i)–(x). This newly created FF database for the CR covering the 2001–2023 period was utilized for all statistical analyses presented in this paper.

### 3.2 Statistical analysis

The database facilitated the identification of individual FF events and days with FF occurrences. An individual FF event (FFE) 160 was defined as a spatially consistent event connected with one locality within a diameter of no more than 35 km affecting one or more municipalities. A day with FF (FFD) was counted as any day during which one or more FFEs occurred. Temporal fluctuations in the frequencies of these variables during 2001–2023 were analyzed, and linear trends were calculated using the non-parametric Theil-Sen method, which offers robustness to outliers in time series (Sen, 1968; Theil, 1992). The non-parametric Mann-Kendall test assessed the statistical significance of linear trends (Mann, 1945; Kendall, 1975). The two-165 proportion Z-test (Sprinthall, 2011) was used to test the significance of differences in relative frequencies of circulation types on days with rainfall totals triggering FFs, and in the corresponding relative frequencies of those types during the entire summer half-year (April–September) in the 2001–2023 period.

### 3.3 Critical point method assessment

The frequent occurrence of severe FFs in the CR led to the development of the 'critical point' methodology. This method 170 identifies small catchments that have a high potential for generating FFs with adverse effects on urbanized areas. A critical point is determined by the intersection of a built-up area's boundary with surface runoff pathways (Drbal et al., 2009). This



approach uses digital elevation model (DEM) data, physiographic parameters, and rainfall characteristics, aggregating them into a critical conditions criterion for each catchment (for more details, see Štěpánková et al., 2017; Drbal et al., 2022). In the CR, over 9,200 critical points were identified, primarily to prioritize activities within (flash) flood risk management
frameworks.

## 4 Results

### 4.1 Spatiotemporal variability of flash floods

A total of 233 individual FFEs were detected in the CR from 2001 to 2023, averaging 10.1 FFEs per year. These events occurred over 160 days, with an average of 7.0 FFDs per year. The highest incidence was in 2009, with 26 FFEs over 16 days,
followed by 25 FFEs over 17 days in 2014 (Fig. 2ab). In nine other years, the number of FFEs was ≥10. A singular FFE was recorded in 2015, with three in 2017 and five in 2022. Despite significant inter-annual variability, both datasets exhibited zero linear trends when analyzed using the Theil-Sen method. In the annual distribution (Fig. 2c), June had the highest occurrence of FFEs (37.8%) and FFDs (33.1%), followed by July (21.9% and 25.0%, respectively), August (17.2% and 18.8%, respectively), and May (17.6% and 15.6%, respectively). FFEs were also recorded in April and September, exclusively during
the summer half-year (April–September). Figure 2d provides additional information, combining the occurrence months of each FFE with the number of affected municipalities in the CR. While most FFEs affected only 1–2 municipalities in various months, significant events occurred in June 2009, affecting 18 municipalities in eastern Moravia, and in July 2010, affecting 15 municipalities in northern Bohemia.



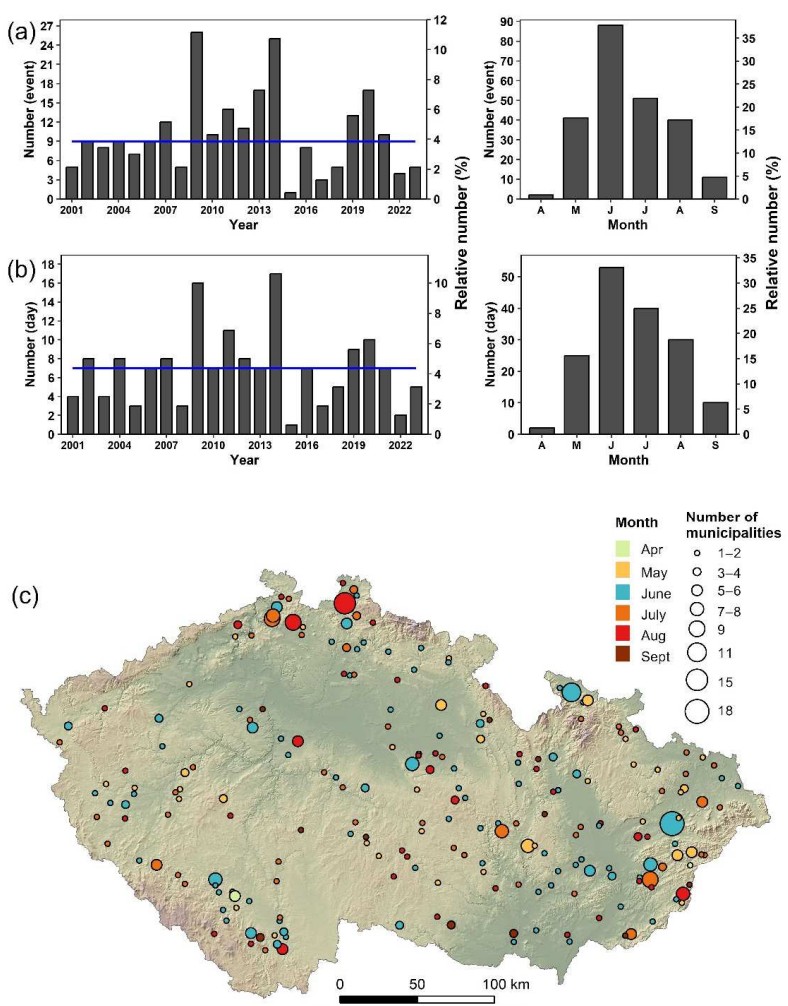

**Figure 2: Flash floods in the Czech Republic in the 2001–2023 period: fluctuations with linear trends (left) and monthly distribution (right) in number of flash flood events FFEs (a) and flash flood days (b); (c) the spatial and monthly distribution of FFEs.**

In total, 501 municipalities in the CR were affected by FFEs during the 2001–2023 period, averaging 2.2 municipalities per individual FFE and 3.1 municipalities per FFD. The most catastrophic FFE occurred in 2009, impacting 86 municipalities (18 on 24 June, 11 on 26 June, and nine on 4 July), followed by 62 municipalities in 2014 (nine on 31 July) and 46 in 2010 (15 on 7 July) (Fig. 3a). While FFEs affected 30 or more municipalities in 2013 and 2020, fewer than ten municipalities were impacted in 2001, 2008, 2015 (only one municipality), 2017–2018, and 2022–2023. A single municipality





was affected during 143 FFEs (61.4%), two in 36 (15.5%), and three in 20 events (8.6%). The annual number of affected municipalities showed a slightly decreasing linear trend (–2.4 municipalities per decade), but this was not statistically significant. Since some municipalities were affected more than once, Fig. 3b illustrates the spatial distribution of all affected

municipalities, represented by the number 424. Prague, Srbská Kamenice, Višňová, and Zlín were hit four times; Česká Ves, Hřensko, Janská, Karolinka, and Vsetín three times; and another 55 municipalities twice. A higher concentration of affected municipalities was particularly noted in northern Bohemia, southwestern Bohemia, and eastern Moravia, as well as in many other smaller areas.

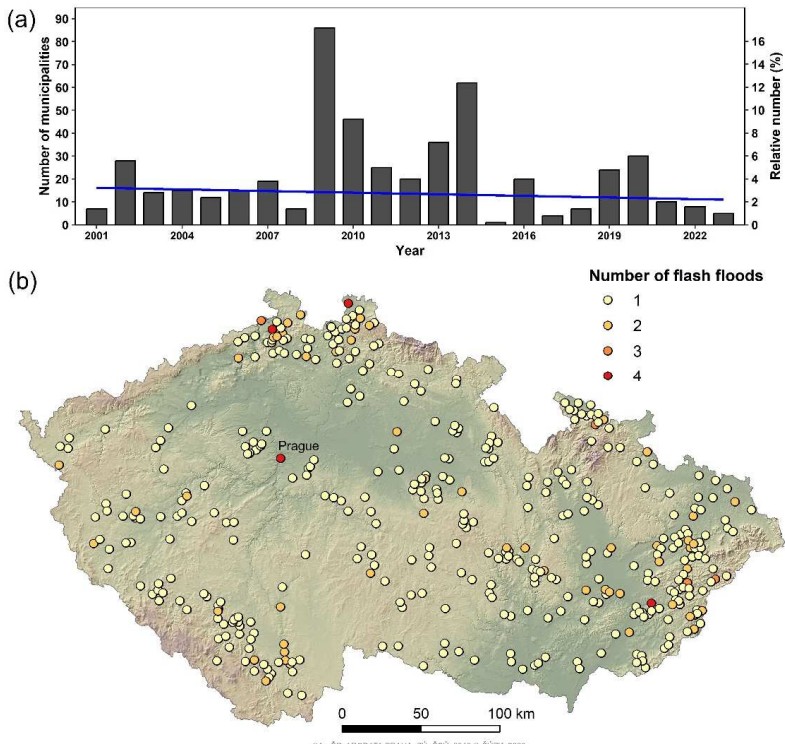

**Figure 3: Municipalities affected by flash floods in the Czech Republic during the 2001–2023 period: (a) fluctuations and linear trends in annual numbers, (b) the spatial distribution.**



### 4.2 Triggering factors of flash floods

### 4.2.1 Meteorological factors

The highest 1-hour, 2-hour, 3-hour, and 24-hour (daily) precipitation totals at the stations of the CHMI network located directly

in the affected area or nearby were represented for all FFEs as box-plots, considering also thunderstorm occurrences (Fig. 4a). The median of these totals (consistently lower than the mean) increased from 20.0 mm (1 hour) to 25.5 mm (2 hours) and 29.3 mm (3 hours), with absolute maxima of 88.3 mm (9–10 Central European Time, CET), 115.4 mm (9–11 CET), and 118.1 mm (9–12 CET) recorded on 14 June 2020 at the Konárovice station (central Bohemia). For daily precipitation totals corresponding to FFEs, the median was 52.0 mm (mean 60.0 mm), with a maximum of 180.5 mm observed on 7 August 2002 at the Pohorská

Ves station (southern Bohemia). Totals were higher across all characteristics for events with thunderstorms than those without. Based of Mann-Whitney U test (Mann and Whitney, 1947), differences in mean 1-hour, 2-hour, 3-hour and daily rainfall totals for stations with observed thunderstorms were significantly higher than for stations without this phenomenon ($p < 0.05$ for daily totals and $p < 0.01$ for others). The results were affected by the proximity of CHMI stations with the highest totals to the core flash flood area (Fig. 4b). For hourly totals, the mean distance was 10.5 km, with a maximum of 25.8 km, whereas for

daily totals, the corresponding distances were 8.6 km and 23.8 km, respectively. Moreover, the dataset comprised 149 stations providing hourly data, fewer than the 203 stations reporting daily totals (selected from a total of 349 and 799 CHMI stations, respectively).

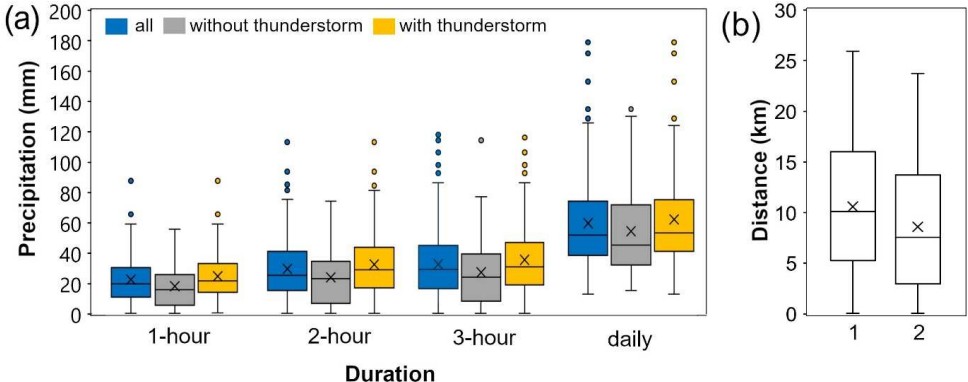

**Figure 4: (a) Box-plots (median, upper and lower quartile, maximum and minimum, outliers; x – mean) of the highest precipitation**
**totals (1-hour, 2-hour, 3-hour, and daily for all cases, without and with thunderstorm occurrence) during flash flood events in the Czech Republic from 2001 to 2023 and (b) box-plots of minimum distances of the core flash flood area from related CHMI stations measuring the highest hourly (1) and daily (2) totals.**

To analyze circulation patterns on days with rainfall totals triggering FFs in the CR, individual circulation types from the objective classification (Sect. 2.2) were assigned to each FFD. As shown in Fig. 5, triggering totals predominantly occurred

during the central cyclone type C on 21 days (13.0% of all 162 analyzed days), followed by the eastern directional type E on



18 days (11.1%) and the cyclonic northeastern type CNE on 11 days (6.8%). Regarding groups of circulation types, triggering rainfall occurred on 69 days during cyclonic types (42.6%, compared to an average frequency of 17.9% in April–September from 2001 to 2023), on 53 days during directional types (32.7% compared to 30.4%), and on 26 days during anticyclonic types (16.0% compared to 48.8%). In total, 14 days were classified under the unclassified type U (8.6%, compared to just 3.0%).

Types C, CN, CNE, CE, CS, E, and U, along with the cyclonic group, showed significantly higher relative occurrences on days with triggering rainfall totals than in April–September, while for the 3 anticyclonic types (A, ASE, AW) and the entire anticyclonic group, the opposite was true.

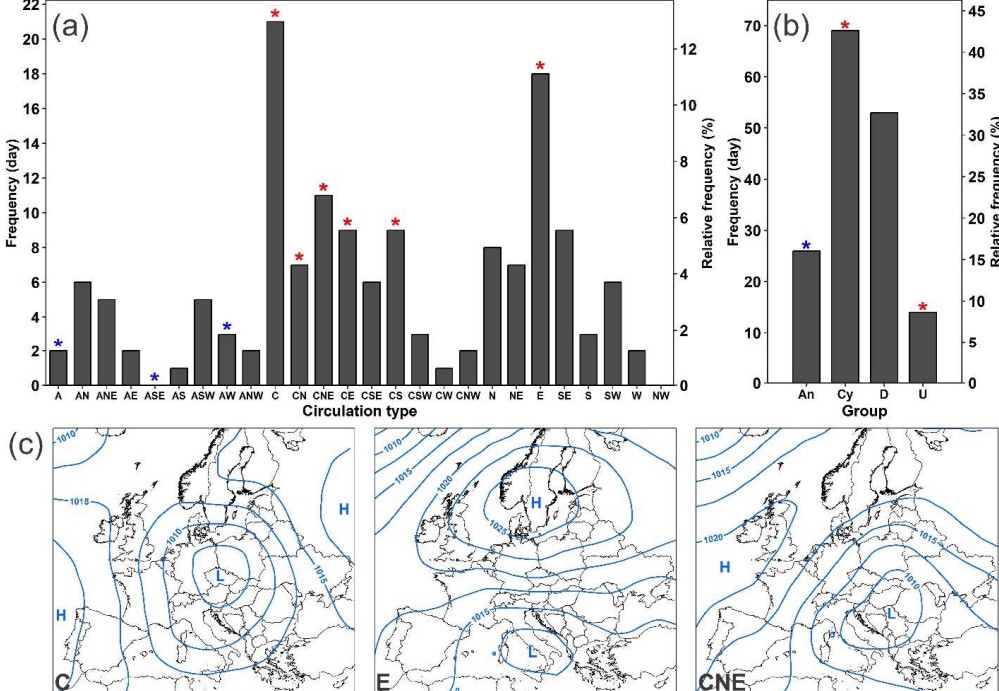

**Figure 5: Frequencies (days/%) of (a) individual circulation types and (b) groups of circulation types (An – anticyclonic, Cy –**
**cyclonic, D – directional, U – unclassified) of the objective classification in days with rainfall totals triggering flash floods in the Czech Republic in the 2001–2023 period; (c) mean sea level pressure fields (H – high, L – low) for the three most frequent circulation types (C – central cyclone, E – directional eastern, and CNE – cyclonic northeastern). The symbol * indicates types with statistically significant (p < 0.05) positive (red) and negative (blue) differences between mean relative frequencies of circulation types on rainfall-rich days and all days from April to September in the 2001–2023 period.**

**4.2.2 Hydrological and geographic factors**

Though intense rainfalls in the CR are somewhat randomly distributed, FFEs predominantly occur at the foothills of the Šumava Mts., Jizerské hory Mts, Frýdlantská pahorkatina Hilly Land, Rychlebské hory Mts., Jeseníky Mts., and Moravian-





Silesian Beskids, as well as across the Bohemian-Moravian Highlands (Fig. 6). These events are most common in catchments with physiographic parameters conducive to transforming causative rainfall into concentrated surface runoff. According to
Drbal et al. (2009), key factors include catchment size, land use, average slope, and relief fragmentation, which influence thalwegs and river network characteristics. Flash flood clusters are notably pronounced in northern Bohemia, especially in the Labské pískovce Sandstones and Jizerské hory Mts. In the eastern CR, areas of concern include the Moravian-Silesian Beskids, Západobeskydské podhůří Foothills, Hostýnsko-Vsetínská hornatina Mts., and Vizovická vrchovina Highlands, where topographic prominence and lithology (flysh) play a significant role.

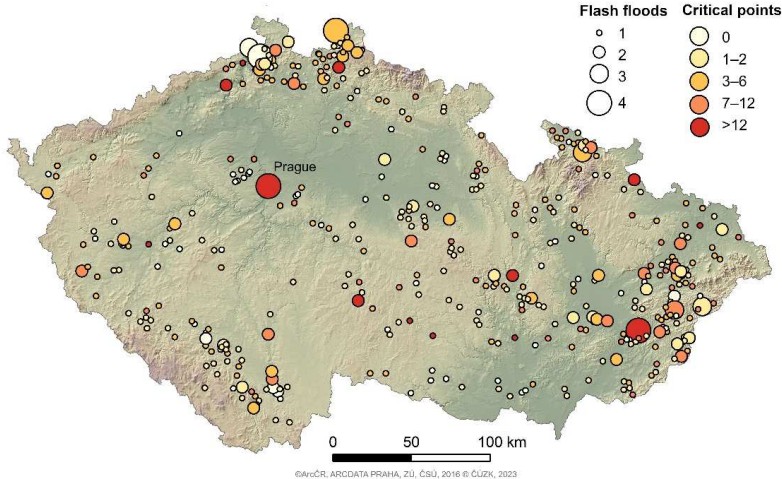


**Figure 6: Spatial distribution of municipalities affected by flash floods in the Czech Republic from 2001 to 2023, highlighting the repeated occurrence in certain municipalities alongside the number of critical points in municipal cadastres.**

Figure 6 also details the number of critical points in municipalities with FFEs over the past 23 years. Prague leads with 25 critical points, followed by other major cities like Zlín (23 points), Ostrava (22), Plzeň (20), Liberec (18), and both
Ústí nad Labem and Brno (17 each, like Krnov). Out of 424 affected municipalities, 62 (14.6%) had no critical points identified, either due to non-conforming catchment criteria or because they experienced extreme rainfall events.

The majority of events in our FF database occurred in hydrologically unobserved profiles, preventing systematic recurrence interval or flood magnitude assessments. While information about watercourses was available for 53.1% of FFEs, only 25.6% had hydrological data. Recorded discharges varied, with recurrence intervals from 2–5 years ($Q_{2-5}$) to 1000 years
($Q_{1000}$). A notable record was on 30 June 2006 for the Dyje River at Podhradí nad Dyjí (southern Moravia), where discharge escalated from 10.4 $m^3s^{-1}$ to 551 $m^3s^{-1}$ (>$Q_{1000}$) within 24 hours (Soukalová et al., 2006). Another significant event occurred on 24 June 2009 at the Jičínka watercourse (northern Moravia), where the water level in Nový Jičín surged to 5 m in two hours,





with a peak discharge of 340 m³s⁻¹ (approximately $Q_{500}$), compared to the mean annual discharge of just 0.81 m³s⁻¹ for this profile (Kubát, 2009).

**4.3 Human impacts of flash floods**

The recorded FFs resulted in material damage and loss of life in the affected areas. Regarding material damage, classified into eight categories (Sect. 3.1) in the CR from 2001 to 2023 (Fig. 7a), the most common category was C (flooded streets/roads), constituting 24.3% of all damage reports, followed by B (flooded houses – 21.7%), A (flooded cellars/basements – 18.3%), D (flooded gardens – 12.3%), and E (damaged roads – 10.5%). Category F (other damage), accounting for 3.7% of records,

included diverse incidents like water well pollution, damaged pond dam, animal deaths, playground destruction, etc. General damage reports (category H) without specific details made up 8.1% of all cases. Landslides (category G) were rare, noted in only 1.1% of incidents. Although some FFEs have preliminary financial damage estimates, the lack of comprehensive data prevents a thorough analysis.

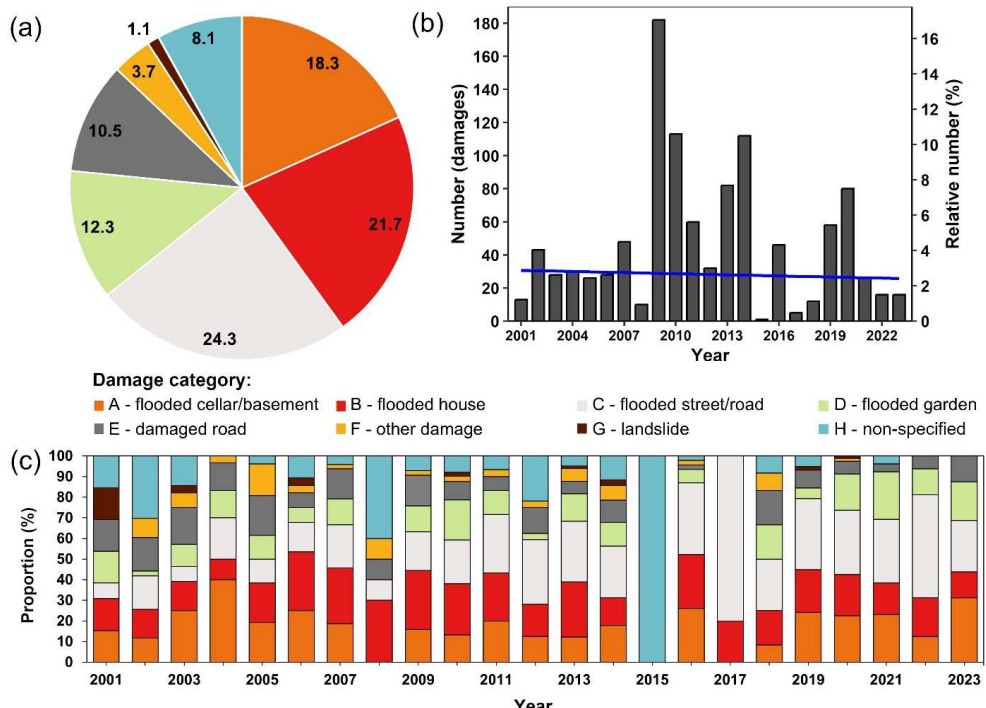

**Figure 7: Damage categories caused by flash floods in the Czech Republic from 2001 to 2023: (a) relative proportions (%) of damage categories over the entire period; (b) fluctuations and linear trend in the absolute annual number of damages; (c) annual relative proportions (%) of individual damage categories.**



The annual fluctuation in reported damages (Fig. 7b) mirrors the inter-annual variability in FFEs (see Fig. 2a) and affected municipalities (see Fig. 3a). The year 2009 had the highest annual tally of all damages (17.1% of the total), followed by 2010 (10.6%), 2014 (10.5%), 2013 (7.7%), and 2020 (7.5%). Conversely, the years with the fewest reported damages were 2015 (one incident), 2017 (five incidents), 2008 (10 incidents), 2018 (12 incidents), and 2001 (13 incidents), collectively accounting for only 3.8%. The annual damage count showed a statistically insignificant, slight downward trend (–2.2 damages per decade) (Fig. 7b). The most prevalent damage type, C (flooded street/road), had annual peaks in 11 years (Fig. 7c), while categories B (flooded house), A (flooded cellar/basement), and H (non-specified damage) were less frequently the most common in any given year, peaking in four (B) and three years (A, H), respectively.

FFEs in the CR during 2001–2023 were accompanied by 36 fatalities, of which 27 (75.0%) were identified as direct and 9 (25.0%) as indirect victims. Direct deaths were associated with torrential water flows causing buildings to collapse, rapid flooding of houses, or sweeping people away, as well as cases where individuals attempted to assist others or save property. Indirect deaths were due to heart attacks triggered by stress from the disaster, delayed arrival of emergency services to people in need because of impassable roads, or hypothermia due to prolonged exposure in water. The behavior of the majority of fatalities can be classified as non-hazardous. However, in a few cases, individuals were too close to flooded watercourses (*e.g*., to observe high water or for unknown reasons) when a bank suddenly collapsed, leading to drowning. Males accounted for 61.1% of all fatalities, and compared to female fatalities (38.9%), they were more frequent in the age categories between 20 and 59 years. Male and female deaths in the age interval of 60–69 years were identical, while for ages ≥70, female fatalities were more frequent. Regarding individual years, 15 people (41.7%) died during FFEs in 2009, and nine (25.0%) in 2010. Three fatalities were recorded in 2002 and 2020 (8.3% each), two (5.5%) in 2018 (a 27-year-old woman and a 28-year-old man during geocaching in Prague), and one in the other four years (2.8% each).

The human impacts described above underscore the importance of spatiotemporal information on municipalities affected by FFEs within the 14 individual administrative regions (*kraj*) of the CR for regional risk management. Table 1 provides an annual overview for the 2001–2023 period. In total, 70 municipalities were affected by FFs in the Zlín region, followed by 54 in the South Bohemian region, 52 in the Olomouc region, 49 in the Liberec region, and 48 in the South Moravian region. Conversely, FFEs affected Prague 4 times and only 6 municipalities in the Karlovy Vary region. The highest annual numbers of 27 affected municipalities were recorded in 2010 in the Liberec region and 19 in 2009 in the South-Bohemian region, followed by 18 municipalities in 2009 in the Ústí nad Labem region and in 2014 in the Zlín region. When the number of affected municipalities in individual regions was related to their total numbers (Fig. 8), the Liberec (18.6% of all its municipalities) and Zlín (17.9%) regions were relatively the most affected, while the Karlovy Vary (3.8%), Vysočina (3.7%), and Central Bohemian (2.6%) regions were the least affected.



**Table 1: The annual totals of municipalities affected by flash floods in the individual administrative regions of the Czech Republic during the 2001–2023 period: PRR – Prague, the Capital City Region; SBR – South Bohemian Region; SMR – South Moravian Region; KVR – Karlovy Vary Region; VYR – Vysočina Region; HKR – Hradec Králové Region; LIR – Liberec Region; MSE – Moravian-Silesian Region; OLR – Olomouc Region; PAR – Pardubice region; PLR – Plzeň Region; CBR – Central Bohemian Region; ULR – Ústí nad Labem Region; ZLR – Zlín Region. Numbers in bold represent years with maximum numbers of affected municipalities in the given region.**

| Region | 01 | 02 | 03 | 04 | 05 | 06 | 07 | 08 | 09 | 10 | 11 | 12 | 13 | 14 | 15 | 16 | 17 | 18 | 19 | 20 | 21 | 22 | 23 | Total |
|---|---|---|---|---|---|---|---|---|---|---|---|---|---|---|---|---|---|---|---|---|---|---|---|---|
| PRR |  | 1 |  |  |  |  | 1 |  |  |  |  |  |  |  |  |  | 1 | 1 |  |  |  |  |  | 4 |
| SBR | 1 | 9 |  |  | 1 | 5 |  | 3 | **19** |  | 1 | 4 |  | 11 |  | 1 |  | 1 | 2 | 1 |  | 6 |  | 54 |
| SMR |  | 7 | 8 | 3 |  | 2 | 2 |  | 4 | 1 | 1 | 1 |  | **11** |  | 1 | 1 |  | 2 |  | 2 |  | 2 | 48 |
| KVR |  |  |  |  |  |  |  |  |  |  |  |  | **4** |  |  |  |  |  |  |  | 1 |  | 1 | 6 |
| VYR |  | 1 | 1 | 3 | **5** |  |  | 1 | 3 | 1 | 3 | 1 | 1 | 4 |  | 1 |  | 1 | 1 | 1 |  |  |  | 28 |
| HKR |  |  | 2 |  | 1 |  | 1 | 2 |  |  | 2 |  | 3 | 3 |  | **5** |  |  | 3 |  | 1 |  |  | 23 |
| LIR | 1 |  |  |  |  |  | 1 |  | 6 | **27** | 1 | 3 | 6 |  |  | 1 |  |  | 2 | 1 |  |  |  | 49 |
| MSR |  |  | 1 | 2 |  | 2 | 1 | 1 | **12** |  | 1 |  | 1 | 11 |  | 1 |  | 1 | 2 | 3 |  |  |  | 39 |
| OLR |  |  | 1 | 3 |  | 1 | 2 |  | **16** |  | 4 |  | 1 | 8 |  |  | 2 |  | 6 | 5 | 1 |  | 2 | 52 |
| PAR |  |  | 1 |  | 2 | 2 |  |  |  | 4 |  |  |  | 4 | 1 |  |  |  |  | **9** | 2 |  |  | 25 |
| PLR | 2 | 2 |  | 2 | 3 | 1 | 2 |  | 2 |  | 3 | **7** | 7 | 2 |  |  |  |  | 2 | 2 |  |  |  | 37 |
| CBR | 2 | 6 |  | 2 |  |  | 1 |  | 1 |  |  | 2 | **8** |  |  |  |  | 3 | 1 | 2 |  | 1 |  | 29 |
| ULR |  |  |  |  |  |  |  |  | **18** | 13 |  | 2 | 3 | 1 |  |  |  |  |  |  |  |  |  | 37 |
| ZLR | 1 | 2 |  |  |  | 2 | 8 |  | 5 |  | 9 |  | 2 | **18** |  | 10 |  |  | 10 | 3 |  |  |  | 70 |
| Total | 7 | 28 | 14 | 15 | 12 | 15 | 19 | 7 | 86 | 46 | 25 | 20 | 36 | 62 | 1 | 20 | 4 | 7 | 24 | 30 | 10 | 8 | 5 | 501 |

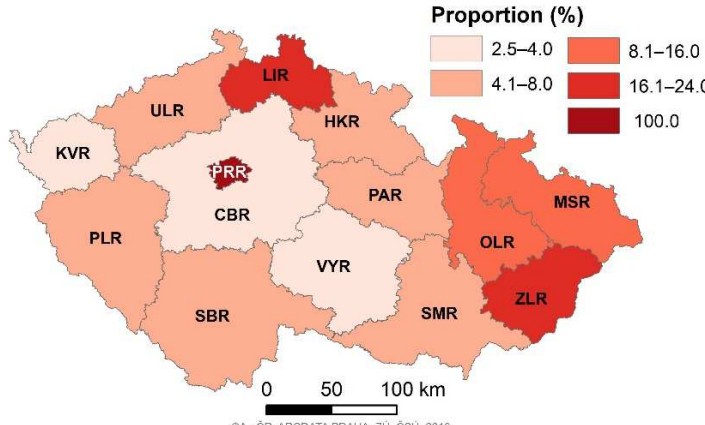

**Figure 8: The relative proportion (%) of municipalities affected by flash floods on their total numbers in the individual administrative regions of the Czech Republic during the 2001–2023 period. For abbreviations of administrative regions see Table 1.**





### 4.4 Significant flash flood events

The three examples of significant FFEs were chosen based on their extremely high peak discharges and precipitation totals to
provide a detailed examination from meteorological, hydrological, and consequential perspectives. Figures 9–11 illustrate the affected areas, highlighting the impacted watercourses, locations, and rainfall totals derived from CHMI rain-gauge and radar observations.

### 4.4.1 15 July 2002

This FFE occurred at the confluence of the Vysočina, Pardubice, and South Moravian regions (Fig. 9a). Intense rainfall in the
area was associated with thunderstorm activity between approximately 4 and 8 p.m., leading to a daily total of 171.7 mm (46.5 mm in one hour) recorded at the Olešnice station, while the local non-CHMI Crhov station reported up to 192 mm (Fig. 9b). The extreme precipitation coincided with the eastern circulation type E, persisting since 13 July and characterized by a gradual increase in cyclonality in the sea-level pressure field over central Europe. Affected streams included Crhovský potok, Dvorský potok, Hodonínka, Loucký potok, and Veselský potok in the Svratka catchment, as well as Petrůvka, Sebránek, Sychotínský
potok, and Úmoří in the Svitava catchment. The most affected Hodonínka stream experienced a peak discharge of $Q_{max} = 110$ $m^3s^{-1}$, indicative of a recurrence interval exceeding 200 years at Hodonín and Štěpánov nad Svratkou. Similarly, high discharges of $>Q_{200}$ were observed at Veselský potok in Olešnice ($Q_{max} = 27$ $m^3s^{-1}$) and Crhovský potok in Crhov ($Q_{max} = 42$ $m^3s^{-1}$), with significant events $>Q_{50}$ also on Petrůvka and Úmoří (see Soukalová, 2002 for more details). The most impacted municipalities were Crhov, Hodonín, Křtěnov, Kunštát, Louka, Olešnice, Štěpánov nad Svratkou, and Zbraslavec. Notably,
Crhov experienced substantial damage, with a road and half of the 30 houses affected, while Olešnice saw damage to five houses, a swimming pool, a cultural center, and a wastewater treatment plant. Tragically, the FF event resulted in two fatalities: a woman in Crhov who died attempting to save her property and another in Hodonín who was trying to escape the floodwaters on the roof of her cottage.



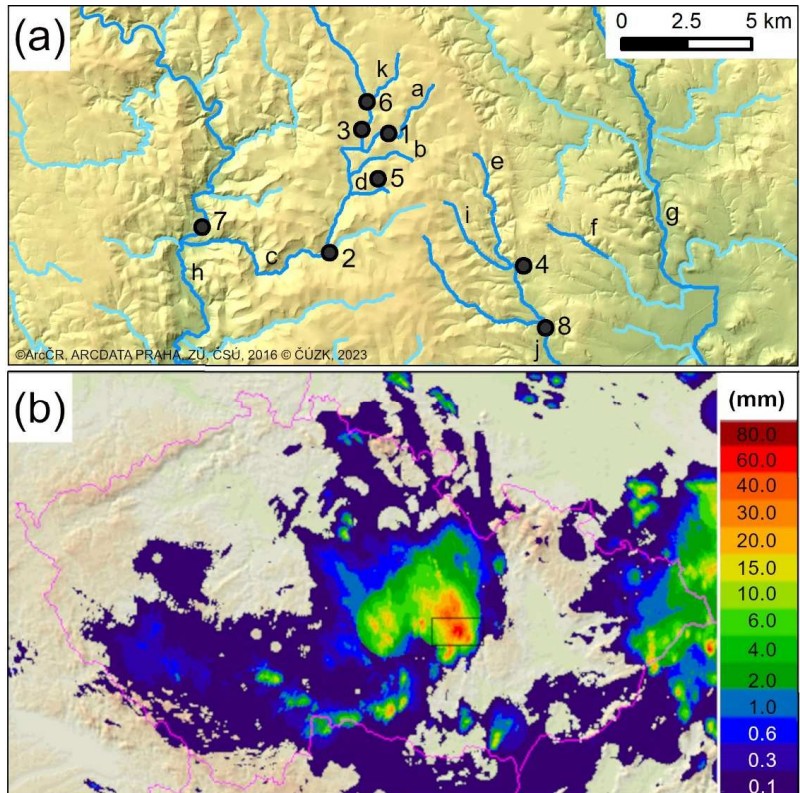

**Figure 9: Flash flood on 15 July 2002: (a) the geographic situation in the affected area, (b) six-hour precipitation totals based on merged radar data across the Czech Republic (the "□" symbol marks the impacted area). Localities: 1 – Crhov, 2 – Hodonín, 3 – Křtěnov, 4 – Kunštát, 5 – Louka, 6 – Olešnice, 7 – Štěpánov nad Svratkou, 8 – Zbraslavec. Watecourses (dark blue): a – Crhovský potok, b – Dvorský potok, c – Hodonínka, d – Loucký potok, e – Petrůvka, f – Sebránek, g – Svitava, h – Svratka, i – Sychotínský potok, j – Úmoří, k – Veselský potok.**

### 4.4.2 24 June 2009

The FFE in the Nový Jičín district and the Moravian-Silesian Region (Fig. 10a) was triggered by torrential rainfall associated with thunderstorms that developed in the afternoon and evening of 24 June 2009. The measured daily total reached 123.8 mm at the Bělotín station (with most rainfall occurring over 2–3 hours), followed by Hodslavice with 120.2 mm and Mořkov with 104.5 mm (Fig. 10b). The event occurred on the third consecutive day of the northeastern directional circulation type NE, succeeded by the type CE, indicating an (north)easterly transport of air masses with an increasing cyclonality of such situations. Some of the most affected streams, like the Bartošovický potok, Grasmanka, Jičínka, Lichnovský potok, Luha, Papakův potok, Sedlnice, Tichávka, and Zrzávka, had $Q_{max} > Q_{100}$, as illustrated by the estimated peak discharges on the Papakův potok





(Mořkov: $Q_{max}$ = 26.8 m³s⁻¹, $Q_{100}$ = 18.5 m³s⁻¹) and Zrzávka (Bludovice: $Q_{max}$ = 135 m³s⁻¹, $Q_{100}$ = 69.5 m³s⁻¹) (Kubát, 2009).
Hodslavice, Jeseník nad Odrou, Kunín, Nový Jičín, Šenov u Nového Jičína, and Životice u Nového Jičína were among the
most affected of the 18 municipalities. Damage primarily impacted infrastructure and residences: 450 houses in Nový Jičín,
110 in Jeseník nad Odrou, 300 in Hranice, 70 in Hustopeče nad Bečvou, and 50 in Bělotín were damaged (*Právo*, 27–28 June
2009, p. 2). In total, this FFE resulted in six fatalities: a woman in Nový Jičín and a man in Životice u Nového Jičína were
swept away by torrential waters, two brothers died while attempting to save their mother, an elderly woman succumbed to a
heart attack after prolonged exposure to cold water in Jeseník nad Odrou, and a man in Kunín died when medical help could
365    not reach him due to the floods. This event was one of several FFEs that occurred in the CR during June and July 2009 (Kubát,
2009), causing material damage of approximately 8.4 billion Czech crowns (around 336 million Euro) and claiming a total of
15 lives.

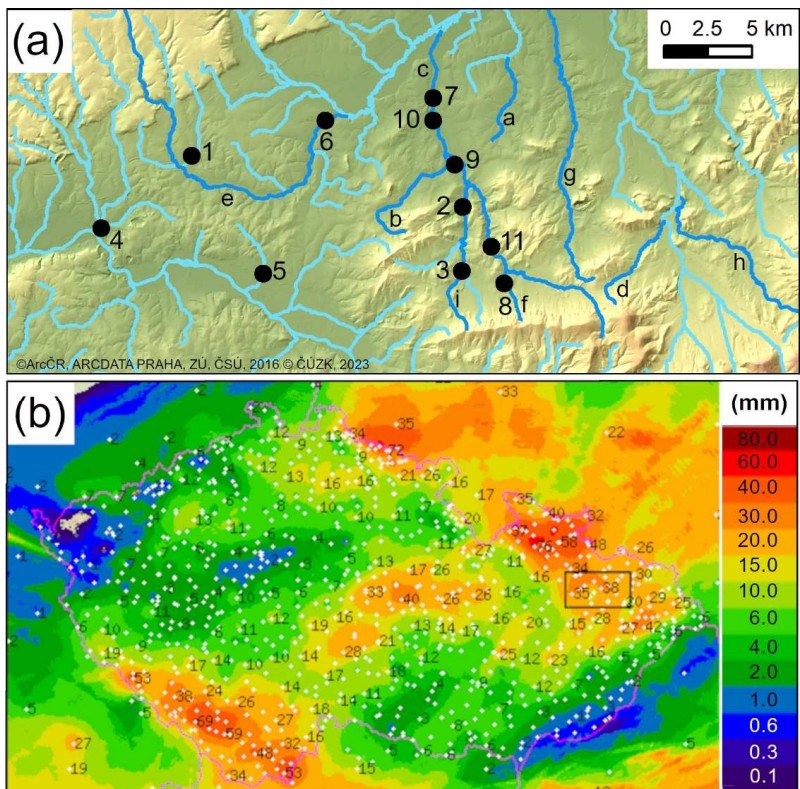

**Figure 10: Flash flood on 24 June 2009: (a) the geographic situation in the affected area, (b) daily precipitation totals according to**
**merged radar information over the Czech Republic (the "▢" symbol marks the impacted area). Localities: 1 – Bělotín, 2 – Bludovice,**



**3 – Hodslavice, 4 – Hranice, 5 – Hustopeče nad Bečvou, 6 – Jeseník nad Odrou, 7 – Kunín, 8 – Mořkov, 9 – Nový Jičín, 10 – Šenov u Nového Jičína, 11 – Životice u Nového Jičína. Watercourses (dark blue): a – Bartošovický potok, b – Grasmanka, c – Jičínka, d – Lichnovský potok, e – Luha, f – Papakův potok, g – Sedlnice, h – Tichávka, i – Zrzávka.**

### 4.4.3 7 August 2010

This FFE occurred in the Liberec region in northern Bohemia (Fig. 11a), where local authorities declared "a state of danger." The daily rainfall on 7 August reached 179.0 mm at the Hejnice station (57.6 mm per 2h, 107.0 mm per 3h), while a local non-CHMI station in Frýdlant recorded a daily total of 190 mm (Fig. 11b). High totals were observed during the northern directional circulation type N, which followed the type NE. The windward effect of the northern slopes of the Jizerské hory Mts. in that region likely enhanced and sustained the rainfall. The main affected streams included the Jeřice, Lužická Nisa, Oleška,

Rynoltický potok, Rokytka, Smědá, Vítkovský potok, and Václavický potok. The Smědá River reached a $Q_{max} = 395$ m$^3$s$^{-1}$, corresponding to a recurrence interval of over 100 years (> $Q_{100}$). At Višňové on the same river, the peak water level was $H_{max}$ = 541 cm, surpassing the extreme flood level of 483 cm for this profile. The most affected municipalities were Bílý Kostel nad Nisou, Dětřichov, Frýdlant, Heřmanice, Chrastava, Raspenava, Hrádek nad Nisou, and Višňové, all experiencing $Q_{max} > Q_{100}$ (Kubát, 2010). For instance, in Heřmanice, all houses were inundated and damaged, with roads and bridges destroyed, leading

residents to await rescue on rooftops (Bernáthová, 2020). In Frýdlant, firefighters evacuated around two thousand people. Total damages exceeded 8 billion Czech crowns (approximately 320 million Euro). The event claimed eight lives, including two men in Raspenava, one in Dětřichov, one woman each in Frýdlant and Heřmanice (the latter swept away with her bed by the water torrent). Moreover, one women died in the Polish town of Bogatynia (near the Czech border), and three persons in the German town of Neukirchen in Saxony (Brumfield and Mortensen, 2010).





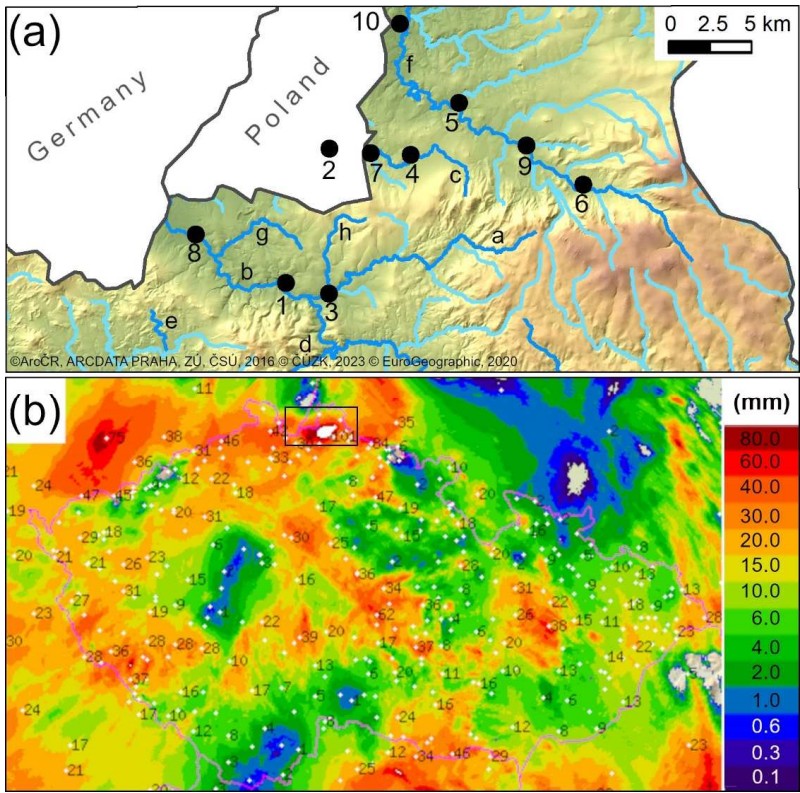

**Figure 11: Flash flood on 7 August 2010: (a) the geographic situation in the affected area, (b) daily precipitation totals according to merged radar information over the Czech Republic (the "□" symbol marks the impacted area). Localities: 1 – Bílý Kostel nad Nisou, 2 – Bohatynia (Poland), 3 – Chrastava, 4 – Dětřichov, 5 – Frýdlant, 6 – Hejnice, 7 – Heřmanice, 8 – Hrádek nad Nisou, 9 – Raspenava, 10 – Višňová, Watercourses (dark blue): a – Jeřice, b – Lužická Nisa, c – Oleška, d – Rokytka, e – Rynoltický potok, f – Smědá, g – Václavický potok, h – Vítkovský potok.**

## 5 Discussion

### 5.1 Data uncertainty

Due to the small territorial extent of FFEs (usually at a local or regional scale), they may remain unnoticed in documentary sources used in this paper. Concerning newspapers, reporting FFEs could have been influenced by several general factors, as specified by Brázdil et al. (2023), such as changes in the space devoted to certain types of information in newspapers, the perceived interest of target readers, the political orientation of the newspaper, the reduction of regional editorial staff, variations in the amount of space given to regional and countrywide reporting, advertising space, competition in reporting, reader fatigue,



and the availability of regional/local news from other sources (*e.g*., police, press agencies, state and regional administration), etc. The use of the internet version of *Novinky.cz* of the *Právo* newspaper either provided the same information or differed only in some details.


Although our database represents the best estimate of FF occurrences across the CR, we must be aware of possible uncertainties, especially in reporting events with small or negligible damage, which could remain unnoticed. These uncertainties can be partly reflected in FF chronologies, their spatial coverage, and descriptions of human impacts. However, such types of uncertainties are more or less a standard feature of databases created particularly from documentary data (see

*e.g*., Brázdil et al., 2006, 2012).

## 5.2 The broader European context

Based on the newly created FF database for the CR in the 2001–2023 period, during the warmest (Zahradníček et al., 2021) and relatively dry (Řehoř et al., 2021b; Brázdil et al., 2022a) phase of the recent warming, FFEs may occur with variable frequency each year (see Sect. 4.1 and Fig. 2), affecting different numbers of municipalities (see Fig. 3). Their occurrence is

concentrated particularly in the summer months of June–August (76.9% of all FFEs and FFDs). When recalculating the number of FFEs per year and area of the CR, corresponding average is 0.13 FFEs/year per 1,000 km$^2$. Comparison with the results of other similar studies is complicated by different approaches to FF definitions and periods analyzed. For example, a higher value of 0.32 FFEs/year per 1,000 km$^2$ for the eastern part of the CR (Moravia and Silesia) is based on events not strongly related to any watercourse and counted for individual municipalities during the 1801–2000 CE period (Halásová and Brázdil,

2020), but applying the same approach, our number increased from 0.13 to 0.28. Slightly higher than our CR number was a corresponding value of 0.19 FFEs/year per 1,000 km$^2$ for the northern and south-western parts of England in the 1700–2013 CE period, including surface water and fluvial flooding (Archer et al., 2019), while for Germany, such values were much lower: 0.02 FFEs/year per 1,000 km$^2$ for FFs that caused damage to any urban area during 1954–2008 (Einfalt et al., 2009) and 0.04 FFEs/year per 1,000 km$^2$ for FF and pluvial flood events in 346–2017 CE, mainly derived from data after 2000

(Kaiser et al., 2021). For another very long period, 1550–2005 CE in Slovenia, Trobec (2017) identified 0.30 FFEs/year per 1,000 km$^2$. Comparable results also come from the Mediterranean, as for example, 0.23 FFEs/year per 1,000 km$^2$ for Catalonia in Spain during 1981–2010 (Llasat et al., 2014), 0.17 FFEs/year per 1,000 km$^2$ for southern France in 1988–2015 (Vinet et al., 2016), or 0.07 FFEs/year per 1,000 km$^2$ for the Campania region in southern Italy during 1540–2015 CE (Vennari et al., 2016).

The initial trigger of each FF is a causative rainfall event and its volume. The resulting runoff is influenced by its

distribution into surface and subsurface runoff, a process determined by the soil's infiltration capacity, hydropedological and hydrogeological characteristics, and land use (Šercl, 2009). The formation of concentrated surface runoff is directly affected by the relief and its geomorphological parameters, which influence the hydrological response of the impacted catchment (Faturová et al., 2024). Orography can also be an amplifying factor in FF generation under certain conditions (Gaume et al., 2009). Additionally, the shape and size of the catchment impact the timing between the occurrence of maximum rainfall

intensity and the peak flow at the outlet (Faturová et al., 2024).



Due to incomplete radar data for studying triggering rainfall that leads to FFEs in the CR during 2001–2023, the study only presented the climatology of such precipitation totals based on the CHMI network, which showed a relatively broad range of totals of different durations, including the occurrence of thunderstorms (see Sect. 4.2.1 and Fig. 4a). However, the results obtained could be biased due to several circumstances concerning the CHMI stations: (i) the rain-gauge station was located at

a great distance from the core FF area; (ii) the maximum total was recorded at a more distant station than the closest one; (iii) hourly totals were not always available from stations with the highest daily total; (iv) the maximum hourly total was recorded at a different station than the one with the maximum daily total. For example, on 20 July 2001, 70 mm of precipitation was reported during 2 hours in Zlín (Galík and Libiger, 2001), while the CHMI Vizovice station, located 19 km from Zlín, measured only 19.4 mm. On 24 June 2009, as much as 112 mm was reported in 2 hours in Nový Jičín (*Právo*, 27–28 June 2009, pp. 1–

3) compared to 74.4 mm measured at the same time in Bělotín (CHMI), 15 km away. On 20 July 2020, 30 mm was reported during 20 minutes in Janovice (Plánička, 2020), but the CHMI Lysá hora station, 8 km away, recorded only 15.7 mm in one hour. These examples indicate that Fig. 4a, based on CHMI station data, likely underestimates the totals of triggering rainfall. Interestingly, Llasat et al. (2016), while analyzing the possible relationship between FFs and convective precipitation in Catalonia (Spain), found an increase in FF frequency but not in terms of extreme precipitation. They considered the effects of

increased vulnerability and exposure to floods and changes in land use as potential explanations. Meyer et al. (2022) analyzed atmospheric conditions favorable for extreme precipitation that triggers FFs in central western Europe for 1981–2020. Although they observed significant increases in atmospheric moisture content and instability, they concluded that "there is no single mechanistic path leading from atmospheric conditions to extreme precipitation and subsequently to flash floods."

The analysis of circulation types on days with rainfall totals triggering FFEs in the CR (Sect. 4.2.1) showed

significantly lower frequencies of anticyclonic types and higher frequencies of cyclonic types, particularly the central cyclone type C, which is characterized by the highest daily precipitation totals (*cf.* Řehoř et al., 2021a). The prominence of the eastern airflow direction (types E, CNE, CE) was notable compared to the western and northern cyclonic and directional types, which generally bring more precipitation to the territory of the CR. This was likely linked to warmer and wetter air masses transported to the CR from the eastern Mediterranean and the Black Sea area by cyclonic circulation during E, CNE, and CE types,

increasing the probability of convective rainfall. Regarding the relatively high frequency of the unclassified type U, it encompassed situations with extremely low air pressure gradients (Zahradníček et al., 2022) over central Europe with weak airflow, leading to radiation warming of the boundary layer in the summer half-year. This warming causes instability and potential for convection, and the static nature of the formed convection, with no frontal systems or wind shear to move convective cells, may contribute to locally high rainfall totals. Palarz et al. (2024) analyzed heavy precipitation events of short

duration in Germany from the radar network for 2001–2020 in relation to "*Großwetterlagen* for Reanalyses." They suggested that such events were influenced by a broader spectrum of circulation patterns, not solely cyclonic, including anticyclonic situations with airflow from the south, characterized by high thermal instability, leading to the development of isolated, smaller convective cells not detected by the rain-gauge station network.



Due to the random nature of triggering rainfall, the rapid onset of flood waves, and the fact that flood formation
predominantly occurs in the late evening and night hours, directly measuring the discharge during a given FFE is practically
impossible (see Sect. 4.2.2). The CHMI estimates flood characteristics from after-flood traces or the nature of the flow from
video recordings by residents. The modeled recurrence interval for the 2009 floods was based on the one-dimensional hydraulic
model HEC-RAS and the rainfall-runoff model HEC-HMS (Šercl, 2009). Evaluating flood discharges by CHMI, for instance,
for August 2010 on the Frýdlantská pahorkatina Hilly Land, was challenging due to damage or destruction of hydrological
stations or because water levels and discharges exceeded the existing rating curve for the hydrological profile. This necessitated
the extrapolation of data and construction of a new rating curve. Newspaper authors often provide inaccurate information
regarding recurrence intervals, so the values published in our study are tied to the official hydrological profiles of the CHMI
and their registration sheets. However, uncertainties in the accuracy of the recurrence interval increase with flood magnitude
($\gg Q_{100}$). The volume of runoff resulting from a precipitation event is influenced by its division into surface and subsurface
runoff, affected by the soil's infiltration capacity, hydropedological and hydrogeological characteristics, and land use (Šercl,
2009). Geographic trigger studies confirmed that nationally, a critical factor in FF formation is a sudden change in the slope
at the foothill, where many villages and small cities are situated. In terms of land use, the increase in urban impervious surfaces
boosts the volume of surface runoff (*e.g.*, Ansari et al., 2016; Wei et al., 2018) by raising the proportion of runoff from rainfall
(*e.g.*, Guan et al., 2016) and reducing runoff response time (*e.g.*, Melesse and Wang, 2007; Miller et al., 2014).

The occurrence of FFs is associated with a wide range of damages, such as to buildings, property, communications,
industrial and water infrastructure, gardens, and fields. By reflecting related information in our FF database, it was possible to
analyze the number of affected municipalities and categorize related damages into eight distinct groups, with damages to
communications and houses being the most common (see Sect. 4.3 and Figs. 7–8). Due to uncertainties in this type of
information (see Sect. 5.1), we did not employ or develop any classification of FFEs or index based on damage data, as done
by Schroeder et al. (2016), Shehata and Mizunaga (2018), or Halásová and Brázdil (2020).

Deadly events during FFEs constitute a significant portion of flood-related fatality databases (*e.g.*, Petrucci et al.,
2019; Papagiannaki et al., 2022). Barredo (2007) attributed 40% of all flood-related fatalities in Europe to FFs during the
1950–2006 period. Špitalar et al. (2014), analyzing 21,549 FFEs in the U.S. from 2006 to 2012, identified short flood durations,
small catchment sizes in rural areas, vehicles, and nocturnal events with low visibility as key factors influencing the number
of injuries and fatalities. Terti et al. (2017) reported 1,075 fatalities in the U.S. from 1996 to 2014, averaging 56.6 fatalities
per year, with no clear trend in such events and associated fatalities. Ahmadalipour and Moradkhani (2019) extended the period
to 2017, recording 1,399 fatalities across the contiguous U.S. Vinet et al. (2022) noted a decrease in the average toll of FFs but
an increase in the number of deadly events in French departments around the Mediterranean from 1980 to 2020. Diakakis et
al. (2023) observed a statistically significant rise in the number of fatalities during 132 significant FFs (with at least 10
fatalities) in 13 countries around the eastern Mediterranean from 1882 to 2021. Our new database recorded 36 FF-related
fatalities across the CR from 2001 to 2023 (Sect. 4.3), averaging 1.6 fatalities per year and 0.2 fatality per FFE. These figures
align with previous Czech studies: 39 FF fatalities from 2000 to 2019 (Brázdil et al., 2021), 123 from 1961 to 2020 (Brázdil



et al., 2022b), and 194 from 1921 to 2020 (Brázdil et al., 2023), showing average fatality rates of 1.94, 2.05, and 1.95 per year, respectively, indicating no significant change in fatality rates over time. The deadliest event in the CR in the past 100 years

was the FF on 9 June 1970, with 35 fatalities, 34 of whom were miners in the collapsing lignite mine at Šardice in southeastern Moravia (Cyroň and Kotrnec, 2000). However, an even more tragic FF occurred in the 19th century in the Berounka catchment in western Bohemia on 25–26 May 1872, resulting in at least 240 fatalities (Müller and Kakos, 2004). This event also contributed to the formation of the unique Mladotice landslide lake (Janský, 1976, 1977).

## 6 Conclusions

Based on a comprehensive analysis of FFs in the CR during the 2001–2023 period, the following conclusions can be summarized:

(i) The unique database compiled from documentary sources enabled a detailed study focused on the spatiotemporal variability of FFs, their meteorological, hydrological, and geographic triggers, as well as their human impacts. FFEs exhibited significant inter-annual variability, no linear trend, and an increased concentration in several core areas, although they can occur in any

watercourse across the CR. Recorded FFEs were confined to the period from April to September.

(ii) The standard rain-gauge network is less suitable for determining the climatology of rainfall that triggers FFs, often providing undervalued totals. Radar measurements, which have only been available in a usable form for the last few years, should be utilized instead. The occurrence of thunderstorms was a crucial factor in reaching triggering totals. While triggering totals can occur under different circulation types, the most significant types identified were central cyclone, directional eastern,

cyclonic northeastern, and cyclonic eastern types.

(iii) Although the distribution of triggering rainfall is random, FFEs typically began at the bases of mountain slopes or at places where equally long river reaches or segments of concentrated runoff converge. FFEs were particularly concentrated at the bases of the Šumava Mts., Jizerské hory Mts, Frýdlantská pahorkatina Hilly Land, Rychlebské hory Mts., Jeseníky Mts. and Moravian-Silesian Beskids. The northern and eastern parts of the CR were the most affected.

(iv) The identified FFEs resulted in 36 fatalities, with three-quarters classified as direct victims. Hazardous behavior during the events sometimes led to fatalities. Nearly two-thirds of the damages were categorized as flooded streets/roads, flooded houses, and flooded cellars/basements.

(v) This comprehensive analysis of FFEs in the CR significantly enriches the existing knowledge of these events at central-European or even pan-European levels, where data from the CR were previously only partially considered or overlooked.

There is considerable potential for extending the existing FF database to years before 2001.

(vi) FFEs, with their socioeconomic impacts manifested in human fatalities and material damages to private and public property and infrastructure, pose a significant natural hazard in the CR. This requires attention in flood risk management across various levels of state administration and local government.



*Data availability.* All data used are available at the corresponding author based on a personal request.

*Team list:* Rudolf Brázdil, Dominika Faturová, Monika Šulc Michalková, Jan Řehoř, Martin Caletka, Pavel Zahradníček

*Author contributions.* RB: conceptualization, methodology, formal analysis, data curation, writing – original draft. DF:
methodology, formal analysis, writing – original draft, software, data curation. MŠM: investigation, methodology, formal analysis, writing – original draft. JŘ: investigation, methodology, data curation, writing – original draft. MC: investigation, formal analysis. PZ: investigation, data curation. All authors contributed to the writing of final version.

*Competing interests.* The contact author has declared that none of the authors has any competing interests.


*Acknowledgements.* DF acknowledge support from the Masaryk University (grant number MUNI/A/1469/2023). We are grateful to Laughton Chandler (Charleston, SC) for his assistance with English style corrections.

*Financial support.* This research was supported by the Johannes Amos Comenius Programme (P JAC), project No.
CZ.02.01.01/00/22_008/0004605, Natural and anthropogenic georisks.

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
