# Peer review of "Spatiotemporal variability of flash floods and their human impacts in the Czech Republic during the 2001–2023 period"

_EGUsphere, 2024_

## Author Comment (AC1)

Flash floods, because they are unexpected, have severe consequences and there is not enough time to prepare for their occurrence, are becoming quite a serious societal problem. Therefore, the creation of a database on their occurrence and the negative consequences caused is an essential prerequisite for the systematic management of flood risk in river basins.

I have a few comments on the work.

RESPONSE: We would like to thank Dr. L. Solín for evaluation of our paper and raising several critical comments, which we are trying to answer below.

- *Note on flash flood definition:*

Consider flash floods wihout pluvial flooding (Kaiser et al.2021) is not correct. During flash flood overland flow from a catchment, which occurs when the rainfall intensity is greater than the infiltration capacity of the land surface (pluvial flooding), is the critical component that contributes to a sudden and significant increase in the flow in the river channel and causes the water to overflow out of the channel (fluvial flooding). Flash flood is a mutual combination of fluvial and pluvial flooding.

RESPONSE: There exist a variety of different definitions of flash floods which are well known to the authors of this manuscript. In our study, we use the term "flash flood" as an umbrella term for all floods caused by intense rainfall events, usually of sudden onset and of a short duration, as supported by different definitions, extending those cited in our manuscript. For example, the United Nations Office for Disaster Risk Reduction (UNDRR) definitions are as follows: "A flash flood is a flood of short duration with a relatively high peak discharge in which the time interval between the observable causative event and the flood is less than four to six hours (WMO, 2006). Surface water flooding is that part of the rain which remains on the ground surface during rain and either runs off or infiltrates after the rain ends, not including depression storage (WMO, 2012)." Wheater and Evans (2009), Miller and Hutchins (2017) and Allegri et al. (2024) consider pluvial flooding, that occurs when surface runoff generation exceeds infiltration rates and drainage capacity, often during high-intensity short-duration rainfall events, as a result from the combination of unfavourable hydro-meteorological a geomorphological conditions, including a failure of flood protection structures. A definition by United States' National Weather Service says: "A rapid and extreme flow of high water into a normally dry area, or a rapid water level rise in a stream or creek above a predetermined flood level, beginning within six hours of the causative event (e.g., intense rainfall, dam failure, ice jam). However, the actual time threshold may vary in different parts of the country. On-going flooding can intensify to flash flooding in cases where intense rainfall results in a rapid surge of rising flood waters."

Based on the examples presented, it is apparent that the definitions of the individual types of floods can overlap and specifically can be selected by individual researchers. Although we agree that the flash flood is a mutual combination of fluvial and pluvial flooding, we noted on lines 71–72 that "our study PRIMARILY CONSIDERS FLASH FLOODS DIRECTLY CONNECTED TO A WATERCOURSE, excluding cases associated with torrential rain causing surface runoff from fields, slopes, or streets in settlements".

References:

Allegri, E., Zanetti, M., Torresan, S., and Critto, A.: Pluvial flood risk assessment for 2021–2050 under climate change scenarios in the Metropolitan City of Venice, Science of The Total Environment, 914, 169925, https://doi.org/10.1016/j.scitotenv.2024.169925, 2024.

Miller, J.D. and Hutchins, M.: The impacts of urbanisation and climate change on urban flooding and urban water quality: A review of the evidence concerning the United Kingdom, Journal of Hydrology: Regional Studies, 12, 345–362, https://doi.org/10.1016/j.ejrh.2017.06.006, 2017.

Wheater, H. and Evans, E.: Land use, water management and future flood risk, Land Use Policy, 26, Supplement 1, S251–S264, https://doi.org/10.1016/j.landusepol.2009.08.019, 2009.

National Weather Service. NWS Glossary. https://forecast.weather.gov/glossary.php?word=flash+flood, last access: 2 August 2024.

UNDRR, Sendai Framework Terminology on Disaster Risk Reduction: Flash Flood https://www.undrr.org/understanding-disaster-risk/terminology/hips/mh0006, last access: 2 August 2024)

UNDRR, Sendai Framework Terminology on Disaster Risk Reduction: Surface Water Flooding, https://www.undrr.org/understanding-disaster-risk/terminology/hips/mh0012, last access: 2 August 2024.

WMO: Technical Regulations. Volume III: Hydrology, WMO-No. 49. World Meteorological Organization (WMO), https://library.wmo.int/viewer/35631/download?file=49_III_en.pdf&type=pdf&navigator=1, 2006.

WMO: Definition number 1465. International Glossary of Hydrology. WMO-No. 385. World Meteorological Organization (WMO), https://library.wmo.int/viewer/35589/download?file=wmo_385-2012.pdf&type=pdf&navigator=1, 2012.

- *Note on data collection.*

Data collection on flash floods based on information in newspapers and web portals as the authors note in the discussion is not exhaustive. As a rule, only major, catastrophic events are reported in these sources. Events who do not have a significant socio-economic impact go unnoticed. This type of uncertainty is considered by the authors to be a standard feature in such data collection. That is true, but how to deal with it, the authors do not give an answer. One way to get the most complete information about flash flooding is to analyse the frequency of declarations of Level 3 flood activity, which are usually declared by mayors of municipalities, in relation to meteorological and circulation patterns.

RESPONSE: As we argue in Sect. 2, we used newspapers, internet sources, CHMI data and professional papers. We doubt, that "events who do not have a significant socio-economic impact go unnoticed", because it always depends on more circumstances leading to reporting of a particular event, and potential socio-economic impact need not to be always "significant". Yes, working with this type of evidence in broader spatial scale as the Czech Republic can lead to loss of some information, what we are fairly saying in discussion of uncertainties. To avoid at least partly this problem, we have to use maximum evidence and sources available, which we tried to do in our study. We are sure, that our data are the most comprehensive dataset of flash floods existing in the Czech Republic despite different attempts made such flash flood set there. Moreover, we also have checked all second and third SPA (Level of flood activity) mentioned in CHMI flash flood reports – date and place using the internet to find out if there were any reports about consequences (damages) in municipalities and those with flooding were added into the database, but as we mentioned in 2.1.3 CHMI reports: "They also include instances where water levels in watercourses increased suddenly but did not overflow their banks; such cases were not included in our FF database."

- *Note on hydrological and geographical factors influencing the occurrence of flash floods*

Compared to the meteorological and climatic aspects, the section on the influence of hydrological and geographical factors on spatial variability is treated in a very general way. Only sites with flash floods are listed and shown, and only a general statement is made that

factors such as catchment size, land use, average slope, and relief fragmentation, river network characteristics lithology are considered to be key in terms of their influence on flash floods. However, any analysis of the geographic attributes of the catchments in which flash floods have occurred, or the hydrographic attributes of their watercourses in relation to, for example, the frequency of flash floods, is entirely lacking. A key hydrological characteristic in relation to the occurrence of flash flooding is the base flow index, but this is not mentioned at all.

RESPONSE: Although the influence of hydrological and geographical factors is indisputable, our study is primarily focused on the analysis of the flash flood database and the information that can be extracted from it. Those general statements on the key physiographic factors are based on the findings of research carried out in the Czech Republic in 2009 after the catastrophic flash floods, that occurred in the Luha and Jičínka basins, and on the basis of which the 'critical point' methodology was developed (see the references in Sect. 3.3).

A detailed analysis of the influence of the physiographic factors would go beyond the scope of this study. Our paper would have to be composed of a series of case studies for which both meteorological and hydrological data were available, including the information on the genesis of overland flow in locality. Yet, in the scientific literature, the lack of hydrological data is frequently discussed problem in connection with flash flood-related research. To conclude, we wanted to draw basic general findings to highlight that we are aware that the physiographic parameters play a role in runoff processes leading to the FF occurrence. In the scale of the whole Czech Republic, the performing an analysis of such type would be demanding, but it is one of the potentials of this paper that could be extended in the future work, which we are working on now in the following step (a more detailed study focused on physiographic parameters of catchments with higher values of unit peak discharge).

- *Note to victims of flash floods.*

For the sake of completeness, in the discussion of flash flood victims, it would be appropriate to also mention the victims that occurred in Slovakia. In July 1998, a storm accompanied by strong winds and hailstorms occurred in the basin of the Mala Svinka. In the affected area, more than 100 millimetres fell in about 120 minutes. Fifty Roma from Jarovnice, mainly children, were victims of the torrential wave.

RESPONSE: We agree that the flash flood on the Mala Svinka on 20 July 1998 is one of the largest events on small basins in Slovakia, as mentioned by Bačová Mitková et al. (2018). But we are not commenting individual deadly flash floods outside of the Czech Republic, which also concerns Slovakia. So we are very sorry, but we do not see the relevance of this information for our article.

References:

Bačová Mitková, V., Pekárová, P., Halmová, D., and Miklánek, P.: Reconstruction and post-event analysis of a flash flood in a small ungauged basin: a case study in Slovak territory. Nat Hazards, 92, 741–760, https://doi.org/10.1007/s11069-018-3222-2, 2018.

---

## Author Comment (AC2)

**Remarks to the Editor**
The expertise of the present Reviewer is mainly in the field of historical floods having occurred in past centuries. He is less familiar with documentary data and the published literature on FFs (= Flash Floods) of the present 21st century period. This Reviewer appreciated that the authors of the present submitted manuscript have used the exhaustive data collection made available in the IT-era. The lead author, prof. Dr. Rudolf Brázdil is a world-known expert in historical floods and has published numerous papers in this context.

This Reviewer is a non-paid consultant of the Royal Meteorological Institute of Belgium (RMIB) at Brussels, Belgium. He has published extensively on hydrological catchment modeling, probability distributions of rainfall depths, historical climatology and hydrology.

**Review Procedure**
All co-authors are members of scientific institutions (Masaryk University, Global Change Research Institute of the Czech Academy of Sciences, Masaryk Water Research Institute, Czech Hydrometeorological Institute) located at Brno, Moravia, Czech Republic what indicates a strong coherence in their research activities and in producing the manuscript under review.
RESPONSE: We would like to thank Dr. G. Demarée for evaluation of our paper and raising several critical comments, which we are trying to answer below.

The extensive data base used in this manuscript challenges the scarcity of ready-available information of FFs which are typically events presenting a limited spatiotemporal context. First of all, the printed newspaper *Právo*, the main national newspaper, and the online *Novinky.cz*, were scanned. Did local publications mainly dealing with advertising and providing local events were also taken into account?
RESPONSE: In *Právo* and other main newspapers are usually also some pages concerning of events of particular regions. Moreover, some other newspapers used in this manuscript for data mining (e.g., *Rovnost*) have rather local focus. Moreover, on the internet we found information of many local events (on the level of individual settlements) or even information of local newspapers. But as we mentioned in Sect. 5.1, "Although our database represents the best estimate of FF occurrences across the CR, we must be aware of possible uncertainties, especially in reporting events with small or negligible damage, which could remain unnoticed."

What was the proportion of newspaper data, internet sources, CHMI Reports, professional papers and Other Data Sources? Are there events mentioned in several of these sources at the same time?
RESPONSE: Presented order of FF data sources in the manuscript express also their decreasing quantity, where newspaper and internet version of related newspaper clearly prevailed. Because several events were covered by different sources, quantification of their proportions would not bring any important information for the reader. Despite this, we give some raw data for the referee: For FFEs in the database, 1058 sources appeared repeatedly and 470 uniquely. Each FFE was described from one to seven sources: one source 36% (181), two 35% (177), three 16% (78), four 9% (44), five 3% (15), six 1% (4) and seven 0.5% (2). From these sources *Právo* represented 36%, *Novinky* 26%, CHMI data 9%, *idnes.cz* 7%, *deník.cz* 4%, etc.
Other Data sources were represented by auxiliary meteorological data, which were used for other analyses (precipitation, circulation types).

What are the minor case letters *a* (Rozhovice) and *b* (Nový Jičín) in Figure 1?

RESPONSE: Small letters were used to identify small watercourses as mentioned in the figure caption: "Watercourses: a – Dubanka, b – Jičínka, c – Rychnovský potok." Rozhovice is locality with number 19. The use of different symbols was needed to localize small places/watercourses in the scale of the whole Czech Republic.

Although the frontier region of Šumava Mountains contains many FFs, none is mentioned in figure 1 – is it a coincidence?
RESPONSE: Figure 1 includes only places, watercourses and geomorphological units mentioned directly by their names in the text itself. Although many FFs occurred in the Šumava Mts. region as the referee wrote (see e.g. Figures 2c, 3b, 6), no place or watercourse was mentioned namely in the text.

In the database of Flash Floods (section 3.1) the item 'elevation (in meter above sea level)' was not selected. However, it might not have any interest as events happened maybe mostly in hilly areas. Maybe this factor might show up in the annual totals of FFs in the administrative regions of the CR (see Figure 8).
RESPONSE: Elevation was not explicitly considered in database (Sect. 3.1), because of difficulty to attribute it to any particular object (settlement – to every mentioned, which part of settlement? watercourse – which part of the stream? damaged objects? etc.). It would be extremely difficult to add elevation to administrative regions – mean elevation, elevation of affected places …? As we say on line 246, "FFEs predominantly occur at the foothills" of mountains, where for concentrated runoff are among the key factors rather "average slope, and relief fragmentation" (line 250). On the other hand, we are aware that the physiographic parameters including elevation are important in terms of runoff generation, which was also shown by Faturová et al. (2024 – see manuscript references). In our following work, there will be a focus on a more detailed study of the physiographic parameters of some of those catchments from the FF database (with higher values of unit peak discharge).

This Reviewer would suggest to include the abbreviation FFD in the legend of Figure 2 as was already done with the abbreviation FFE: … and flash flood days (FFDs); …
RESPONSE: Changed as requested: "and flash flood days (FFDs);"

The probability distributions of the FFEs and FFDs in the summer half-year seem to be Gaussian.
RESPONSE: May be yes, but we did not test it, because as we believe this information is not so important for this manuscript.

Line 470: Is there a hydrometeorological reason why the flood formation predominantly occurs in the late evening and in the night? Most probable the temporal occurrence of 'thunderstorms'are a basic argument (seen section 4.2.1).
RESPONSE: We added related information to the following part of the sentence: "… predominantly occurs in the late evening and night hours (following a usual time of the thunderstorm occurrence), directly measuring ..."

Significant FFEs (see section 4.4): no remarks
RESPONSE: Thank you.

Statistical evaluation of deadly events during FFEs is difficult as the reference periods are very different. Some reference periods even include the late 19[th] century which might even question their data base collection techniques. Strictly speaking the rates in a broader

geographical context are only comparable on the time frame of the present data base. However, previous studies by Brázdil, the lead-author, using similar data collection techniques and methodology show clearly no significant change in fatality rates over time in the Czech Republic.

RESPONSE: We are aware of the problem of comparing fatalities for different periods, territories and definitions of FFs. However, our aim was to present not only data from the CR based on preceding studies (which are the best comparable), but rather show different spatiotemporal scales and results of studies dealing with the same topic in many other countries.

Conclusion: This manuscript is clear, well written and has a large exhaustive reference list dealing with FFs. The authors were very successful in exploiting the data bases mentioned in section 2 'Data' in the context of FFE and FFD-related occurrences. Finally, it might be suggested to the authors in a potential future paper to extend the results of the present manuscript dealing with FFs in the Czech Republic in the reference period 2001-2023 to earlier periods.

RESPONSE: Thank for your suggestion. In our study we tried present the first complex analysis of FFs in the Czech Republic in a broader view for the period well covered by different types of data. We already collected a lot of FF information before 2001 and we have been collecting continuously also data from 2024, i.e. there is a hope for an important future extension of the database used for the recent study.

This Reviewer suggests publication as it stands leaving the authors, if they wish, taking care of the few minor suggestions and remarks mentioned by this Reviewer.

RESPONSE: Thank you very much.

---

## Author Comment (AC3)

The work on flash flood is an interesting methodological effort to understand the behavior and evolution of these phenomena. Its interest is growing due to the impacts it causes in a short time and in very limited areas. Hence the complexity of its study and the need to make approximations on a detailed scale and proposing specific methodologies.

RESPONSE: We would like to thank the anonymous referee #3 for evaluation of our paper and raising several critical comments, which we are trying to answer below.

**General comments**

- It would be interesting if the abstract contained a preview of specific results of the research carried out.

RESPONSE: Accepted, abstract was complemented by some numbers as follows (we would not like to repeat other information from conclusions in Sect. 6): "Flash floods, characterized by their sudden onset, extreme discharges, short duration, material damage, and human loss, represent a significant natural hazard. Not well covered by standard hydrological observations, flash floods data can primarily be derived from various types of documentary evidence. This evidence served as the main data source for creating a flash flood database for the Czech Republic from 2001 to 2023. This database enabled detailed analysis of different aspects of flash floods. The annual series of 233 flash flood events, 160 flash flood days, and 424 affected municipalities showed significant inter-annual variability but no linear trends. The triggering rainfalls that generate flash floods were analyzed with respect to 1–3-hourly and daily precipitation totals and circulation types from the objective classification. While flash floods can occur anywhere, they were more frequently recorded at the foots of mountain slopes, often coinciding with "critical points" where built-up areas meet concentrated surface runoff pathways. The division of material damage caused by flash floods into eight categories indicated that the highest proportions of damage were to streets and communications 24.3%, as well as to houses 21.7%, their cellars, and basements 18.3%. There were also 36 recorded fatalities. The understanding of flash floods in the Czech Republic aligns generally well with studies of flash floods in other European regions.

- Would it be possible to define the phenomenon with instrumental or at least quantitative variables? Not only related to hydrological criteria.

RESPONSE: Generally speaking, the flash floods are an unpredictable phenomenon and it is not possible to have some quantitative variable for such number of events. The creation of flash flood depends on diverse variables, the important one is the previous saturation of a catchment and so, the same precipitation can or may not create the surface runoff, the flooding depending on previous conditions. Our research presents the first step in collection and creation of systematic database of these events in the last decades. Of course, we could theoretically use, for example, quantitative precipitation estimates from radar measurements (as the standard meteorological station are not always located directly in or in proximity of impacted area), but such data are not available in the requested quality for the whole period analysed. It means, that for the scale of the whole Czech Republic and the period analysed it is nearly impossible to find some unquestionable instrumental variables.

- Defining flash floods based on hydrological behavior does not completely define flash floods. That is correct, but future research should also consider effects outside the river system. That is, pluvial floods. It's a more complex or diverse approach to be promoted in next steps of research.

In the Mediterranean region, the effects of torrential rain are currently directed in large proportion towards this type of phenomenon, linked to drainage problems, poor urban planning, or to the effect of the great minute intensity in which the precipitation occurs.

Distribution of these events is more extense than fluvial system, affecting areas in which historial or instrumental records cannot describe similar previous situations.
RESPONSE: We agree with your comment. In the future research we can include also the pluvial events. However, pluvial floods are very rare in the geographical conditions of the Czech Republic as the hydrographic network is very dense and any surface runoff soon enters a watercourse, making it very difficult to distinguish between flash and pluvial floods. In other words, a flood, that came through a watercourse does not (have to) contradict the primary cause of the flood – the rapid surface runoff (typically from arable land, built-up areas etc.).

- The work only covers a period of 23 years. It would have been much better to generate robust results for the purposes of climate analysis, to be able to extend the study period, at least up to 30 years.
RESPONSE: We agree with the referee that using the longest period as possible would be good for our research, but we have to reflect also given national/regional conditions for such research as well as the aim of this study. We selected the period which is the best covered by various data sources, particularly of electronic resources, journals web pages, which are in the Czech Republic widely expanded after 2000. This kind of research is extremely time consuming, because it requires a lot of time in searching and reading of different data sources including subsequent data verification and its critics. As one of important contributions of our study we see the creation of a systematic database of flash floods which can be further well extended into the past as well as into the future. Not any other such national database of flash floods exists in the Czech Republic until now.

**Specific comments**
Lines 100-105. Figure 1.
The location map is insufficient. Showing only the study region hinders viewing the entire river systems. A reference to the European continent and Central Europe would be very convenient. Considering foreign readers, it's important aspect to be considered.
RESPONSE: The new version of Figure 1 was added, where your comments (river system, position of the Czech Republic in Europe) were taken into account – see below:

[Figure]

©ArcČR, ARCDATA PRAHA, ZÚ, ČSÚ, 2016 © ČÚZK, 2023 © Moravian National Municipality, 2023

Lines 110 and following
The use of local press sources has positive aspects. But in itself it leaves information unrecorded because it is not systematic and filters the published information by editorial criteria. It should be complemented with reports from emergency management authorities, administrations, insurance companies and social networks where amateur observers generate records of notable quality. Do the authors plan to deepen the research using these more objective and systematic sources of information?

Response: We agree with the referee because we are well aware of the problems with the use of documentary data and its weaknesses, particularly concerning newspaper – see our comments in Sect. 5.1: Data uncertainty. We summarised it on lines 406-407: "Although our database represents the best estimate of FF occurrences across the CR, we must be aware of possible uncertainties, especially in reporting events with small or negligible damage, which could remain unnoticed." We used the best data sources available for our research in the Czech Republic we had available – see detail description in Sect. 2.2 Data of flash floods. There is no problem to try to extend the now existing database from other sources in the future, but there are also other limitations of sources you propose and consider as "more objective and systematic" (e.g., availability of data from insurance agencies and their interest in events with smaller damage, 'non-specified type' events in firemen interventions, etc.).

Regarding the use of information obtained from social networks, of course, its use would only be as a complement as a secondary source, but it can provide a very good level of detail.
RESPONSE: In our research, we check the information from press also by hydrological reports of Czech Hydrometeorological Institute and records from fireman's interventions. In the following research we can complement the reports with social networks.

Lines 210 and following
It would be interesting to explain the availability of precipitation intensity records. Current pluvial flooding situations have a direct relationship with precipitation in millimeters per minute. Minute intensity data, if it exists, would be very explanatory.

RESPONSE: As mentioned in one of the comments above, it is often not possible to unambiguously determine the source areas on which a decisive part of the surface runoff is formed. And thus it is not possible to quantify the causal rainfall total and intensity. It is only possible to present at least the range of recorded 5- or 10-min intensities from weather radar data in the given areas if such data are available, which concerned only part of our period analyzed.

Unfortunately, this flash flood phenomenon "escapes" from the most common records by daily total. For example, in some regions we perceive an increase in flash floods due to episodes that do not reach a large total magnitude. 100 or 200mm of total daily precipitation are not necessary. On the other hand, the effects are very serious due to rain events that reach or exceed 2/4 millimeters per minute for periods of 10 to 20 minutes.

RESPONSE: We agree with the referee that information in mm per minute would be explanatory, but the dataset of CHMI used in this study included 799 stations reporting daily totals and 349 stations (automatic stations) providing hourly data. These automatic stations have even data in shorter time intervals, but such data (i.e. shorter than 1 hour) are not standardly checked for their quality and are biased by some other errors following from automatic measurements itself. Moreover, they do not cover the whole 23-year period analysed, despite the fact that such stations need not to be necessarily located close to the core areas of flash flood occurrences, which could be possibly used in our study. From these reasons our analysis concentrated on uniform high-quality daily and hourly precipitation totals covering the whole 2001–2023 period as presented in Sect. 4.2.1. Moreover, as mentioned in Sect. 2.2, above data were used also with respect "to the limited availability of radar precipitation data", which represent another potential quantitative characteristic of precipitation intensity.

Line 235 and following

Figures 5a and 5b. The data on prevailing winds and weather types in these figures would perhaps be better served with a pie chart style display, or a "compass rose" simulation. Suggested graphic resource is already applied in figure 7a.

RESPONSE: We understand what the referee means, but it is probably some misunderstanding. This data concern circulation types (synoptic situation), not prevailing winds, i.e. expression "with a pie chart style" would not be bringing any new information and would be also probably less instructive as our expression in Figs. 5a and 5b is. From this reason we would like to preserve Figure 5 in the original form.

Line 277.

Concept "preliminary financial damage estimates". It would be better "preliminary economic damage estimates" ?

RESPONSE: Accepted, we changed as proposed.

Line 280. Figure 7c.

This important display of data by categories, it would be better not only a graphic display, but also showing detailed data in a table. To make easier comparative evaluations between concepts and evolution in time. This resource is already used in Table 1.

RESPONSE: Accepted, the following Table was added to the manuscript:

Table: The annual totals of damage categories in municipalities affected by flash floods in the Czech Republic during the 2001–2023 period: A (flooded cellars/basements), B (flooded

houses), C (flooded streets/roads), D (flooded gardens), E (damaged roads), F (other damage), G (landslides), H (non-specified).

| Damage | \multicolumn Year 2000+ | | | | | | | | | | | | | | | | | | | | | | | Total |
|---|---|---|---|---|---|---|---|---|---|---|---|---|---|---|---|---|---|---|---|---|---|---|---|---|
| | 01 | 02 | 03 | 04 | 05 | 06 | 07 | 08 | 09 | 10 | 11 | 12 | 13 | 14 | 15 | 16 | 17 | 18 | 19 | 20 | 21 | 22 | 23 | Total |
| A | 2 | 5 | 7 | 12 | 5 | 7 | 9 | 0 | 29 | 15 | 12 | 4 | 10 | 20 | 0 | 12 | 0 | 1 | 14 | 18 | 6 | 2 | 5 | 195 |
| B | 2 | 6 | 4 | 3 | 5 | 8 | 13 | 3 | 52 | 28 | 14 | 5 | 22 | 15 | 0 | 12 | 1 | 2 | 12 | 16 | 4 | 3 | 2 | 232 |
| C | 1 | 7 | 2 | 6 | 3 | 4 | 10 | 1 | 34 | 24 | 17 | 10 | 24 | 28 | 0 | 16 | 4 | 3 | 20 | 25 | 8 | 8 | 4 | 259 |
| D | 2 | 1 | 3 | 4 | 3 | 2 | 6 | 0 | 23 | 22 | 7 | 1 | 11 | 13 | 0 | 3 | 0 | 2 | 3 | 14 | 6 | 2 | 3 | 131 |
| E | 2 | 7 | 5 | 4 | 5 | 2 | 7 | 1 | 27 | 10 | 4 | 4 | 5 | 12 | 0 | 1 | 0 | 2 | 5 | 5 | 1 | 1 | 2 | 112 |
| F | 0 | 4 | 2 | 1 | 4 | 1 | 1 | 1 | 4 | 3 | 2 | 1 | 5 | 8 | 0 | 1 | 0 | 1 | 0 | 1 | 0 | 0 | 0 | 40 |
| G | 2 | 0 | 1 | 0 | 0 | 1 | 0 | 0 | 0 | 2 | 0 | 0 | 1 | 3 | 0 | 0 | 0 | 0 | 1 | 1 | 0 | 0 | 0 | 12 |
| H | 2 | 13 | 4 | 0 | 1 | 3 | 2 | 4 | 13 | 9 | 4 | 7 | 4 | 13 | 1 | 1 | 0 | 1 | 3 | 0 | 1 | 0 | 0 | 86 |
| Total | 13 | 43 | 28 | 30 | 26 | 28 | 48 | 10 | 182 | 113 | 60 | 32 | 82 | 112 | 1 | 46 | 5 | 12 | 58 | 80 | 26 | 16 | 16 | 1067 |

---

## Author Response (AR2)

**RESPONSES TO EDITOR COMMENTS**

We would like to thank the editor for her comments. We are trying to respond point-by-point below.

**Comment on line 100 (Fig. 1):** I suggest to eliminate locality because it is understandable that the points indicate localities. Moreover, probably this figure could be slightly reduced in size, being a geographical map not containing results of this research.
RESPONSE: Accepted and changed as requested.

**Comment on line 127:** These sources does not focus on damage (as the previous). I suggest to modify the title as for example Meteorological and hydrological data sources.
RESPONSE: Accepted and changed as: "Meteorological data"

**Comment on line 148:** Maybe you can change in "Watercourse/s affected by FF.
RESPONSE: Accepted and changed as requested.

**Comment on line 152:** I suggest to change as: Category of elements flooded: A – cellar/basement, B – house, C – street/road, D – garden, E – road, F – other elements, G – landslide occurrence, H – non-specified element damaged. actually landslide is not homogeneous with the others. In case you accept this suggestion, please update this in the following of the paper.
RESPONSE: We thank for this proposal, but preservation of the original form seems to us as more appropriate. But we changed everywhere "landslide" on "landslide damage" to be more consistent in this context.

**Comment on line 168 ("…in the 2001–2023 period."):** study period
RESPONSE: Accepted and changed as requested.

**Comment on line 173:** Eliminate "data".
RESPONSE: Accepted and changed as requested.

**Comment on line 180:** Eliminate "FFEs".
RESPONSE: This sentence concerns of flash flood days, it is the text "with an average of 7.0 FFDs per year" should remain in the original form

**Comment on line 209:** I suggest to change as: Meteorological framework and rain measurements (or something similar), even if you described first rain and after meteorology.
RESPONSE: Accepted and changed as: "Precipitation and circulation types"

**Comment on line 262 ("…because they experienced extreme rainfall events"):** It sounds strange.
RESPONSE: Accepted and changed as: "or the occurrence of extreme rainfall events"

**Comment on line 272 ("…material damage"):** Change as before.
RESPONSE: It correspond to terminology used in agreement with our response to comment on line 152.

**Comment on line 273 ("…in the CR from 2001 to 2023):** Eliminate.
RESPONSE: Accepted and changed as requested.

**Comment on line 273:** If you accepted my suggestion in 3.1, please change accordingly.
RESPONSE: Please see our response to comment on line 152.

**Comment on line 351 and 375 and 397 (Figs. 9–11):** I suggest to reduce 10% this figure (only the figure, not the legend nor the letters).
RESPONSE: Accepted and changed as requested.

**Comment on line 419 (“…for the CR in the 2001–2023 period”):** Please, find this sentence all around the text that is repeated several times and in some cases change as "study period" to avoid repetitions.
RESPONSE: Accepted and changed as requested.

**Comment on line 424 (“…other”):** Eliminate.
RESPONSE: Accepted and changed as requested.

**Comment on line 443 (“…studying triggering rainfall that leads to FFEs in the CR during 2001–2023”):** Change as for study period.
RESPONSE: Accepted and changed as requested.

**Comment on line 444 (“…climatology of such”):** Eliminate.
RESPONSE: Accepted and changed as requested.